# Secreted EMC10 is upregulated in human obesity and its neutralizing antibody prevents diet-induced obesity in mice

Xuanchun Wang [1,15] ✉, Yanliang Li[1,2,13,15], Guifen Qiang [2,3,15], Kaihua Wang[4,5,15], Jiarong Dai [1], Maximilian McCann [2,13], Marcos D. Munoz[2], Victoria Gil [2], Yifei Yu[1], Shengxian Li[2,6], Zhihong Yang[7,14], Shanshan Xu[2], Jose Cordoba-Chacon [8], Dario F. De Jesus [7], Bei Sun[9], Kuangyang Chen[1], Yahao Wang[1], Xiaoxia Liu[1], Qing Miao[1], Linuo Zhou[1], Renming Hu[1], Qiang Ding[10], Rohit N. Kulkarni [7], Daming Gao [4,11], Matthias Blüher [12] & Chong Wee Liew [2] ✉

Secreted isoform of endoplasmic reticulum membrane complex subunit 10 (scEMC10) is a poorly characterized secreted protein of largely unknown physiological function. Here we demonstrate that scEMC10 is upregulated in people with obesity and is positively associated with insulin resistance. Consistent with a causal role for scEMC10 in obesity, *Emc10⁻/⁻* mice are resistant to diet-induced obesity due to an increase in energy expenditure, while scEMC10 overexpression decreases energy expenditure, thus promoting obesity in mouse. Furthermore, neutralization of circulating scEMC10 using a monoclonal antibody reduces body weight and enhances insulin sensitivity in obese mice. Mechanistically, we provide evidence that scEMC10 can be transported into cells where it binds to the catalytic subunit of PKA and inhibits its stimulatory action on CREB while ablation of EMC10 promotes thermogenesis in adipocytes via activation of the PKA signalling pathway and its downstream targets. Taken together, our data identify scEMC10 as a circulating inhibitor of thermogenesis and a potential therapeutic target for obesity and its cardiometabolic complications.

Obesity is the result of a chronic imbalance between energy intake and expenditure and is a major risk factor for metabolic diseases[1,2] such as type 2 diabetes mellitus, cardiovascular disease, and certain types of cancer[1,3]. Brown and beige fat have recently attracted significant interest as tissues that could be leveraged to treat obesity and diabetes. This is due to their ability to consume a considerable amount of glucose and lipids and dissipate the chemical energy from these substrates as heat, in a process called thermogenesis[4–9]. Mouse models with enhanced brown or beige fat content or activity have been previously demonstrated to resist weight gain and exhibit improved metabolic health via activation of adipose thermogenesis[10–13].

However, despite intense investigation and an increasingly detailed understanding of the molecular determinants of adipocyte thermogenesis in cells and mice, modulation of adipocyte thermogenic capacity via either brown fat activation or increasing the amount of thermogenic adipose tissue has not been successfully implemented as a therapeutic strategy in human obesity. The reasons for this are manifold – but of principle importance is that there are important physiological differences in humans and lower organisms with respect to the regulation and functional relevance of adipocyte thermogenesis that may limit translation. Identifying the determinants of adipocyte thermogenic capacity in humans with obesity

using integrated clinical and pre-clinical studies may identify alternative therapeutic targets for obesity with increased likelihood of clinical translation.

Previously, we identified a secreted protein, scEMC10 (secreted isoform of endoplasmic reticulum membrane complex subunit 10), also known as INM02 and hHSS1[14–16]. EMC10 is highly conserved and has been identified in at least 45 species spanning across classes, phyla, and kingdoms. It is remarkably unique and has no significant homology to any other protein[15]. Differential splicing of the *EMC10* gene produces two EMC10 isoforms – membrane bound EMC10 (mEMC10) and scEMC10. Both isoforms contain a signal peptide and a luminal domain. mEMC10 contains a transmembrane domain at the carboxyterminus and forms a part of the endoplasmic reticulum complex (EMC)[17,18]. scEMC10 lacks a transmembrane domain and consequently, scEMC10 is shuttled into the secretory pathway and can be detected in cell culture media, whereas the canonical isoform is sequestered intracellularly (Supplementary Fig 1)[15,19].

In the last decade, sc*Emc10* was first shown to be upregulated by high glucose in pancreatic β-cells[14]. This secreted isoform was later shown to be highly expressed in high-grade gliomas[15,16], and to promote cardiac tissue repair after myocardial infarction[19]. The membrane bound isoform was found to contribute to schizophrenia[20], and more recently two independent groups observed homozygous *mEMC10* variants were associated with a syndrome of intellectual disability and global developmental delay in humans[21,22]. Additionally, our group showed that EMC10 is indispensable to male fertility via maintaining ion balance in sperm[23]. However, the function of EMC10, especially the secreted isoform, in systemic metabolic regulation has not been explored.

In this work, we use studies in cells, mice, and humans to demonstrate that scEMC10 is a circulating inhibitor of adipose tissue thermogenic capacity and a potential therapeutic target for obesity and its complications.

## Results

### Serum EMC10 levels are correlated with BMI and insulin resistance in humans

scEMC10 is a secreted protein of poorly characterized function. Little to no data exists describing its function in humans and its role in energy balance and glucose homeostasis has not been explored. To address this, we developed a chemiluminescent-based immunoassay for human scEMC10 detection and employed this assay to study the association between scEMC10 and obesity.

We first investigated serum EMC10 levels in a white cohort including human study participants with leanness, overweight, and obesity, and observed that circulating EMC10 levels were significantly upregulated in overweight compared to lean individuals and the upregulation was further exacerbated in patients with obesity (Fig. 1A, Supplementary Table 1). Similar to findings in the white cohort, serum EMC10 levels were higher in human study participants with overweight and obesity than lean controls in a Chinese Han cohort (Fig. 1B, Supplementary Table 2). Regression analysis showed that serum EMC10 levels positively correlated with BMI in both white and Chinese Han cohorts (Fig. 1C, D) and this effect was not altered by sex (Supplementary Fig. 2A–D). These findings were replicated in another white cohort: a weight-loss cohort of human study participants undergoing either bariatric surgery or a combined hypocaloric diet and exercise (Fig. 1E, Supplementary Table 3).

As obesity causes insulin resistance and cardiometabolic disease, we examined the association of serum EMC10 with insulin sensitivity and other cardiometabolic traits. In white human study participants who underwent euglycemic-hyperinsulinemic clamp studies, we observed that serum EMC10 levels inversely correlated with glucose infusion rate (GIR) and positively correlated with fasting plasma insulin levels, demonstrating serum EMC10 covaries with insulin resistance in

humans (Fig. 1F, G). Consistent with an association between serum EMC10 and insulin resistance, serum EMC10 levels correlated positively with fasting plasma glucose, HbA1c, free fatty acids (FFA) and leptin, and inversely with serum adiponectin (Fig. 1H–L). We also observed serum EMC10 levels positively correlated with both subcutaneous and visceral fat areas (Fig. 1M, N). Therefore, serum EMC10 is positively associated with BMI, fat mass, insulin resistance and adverse metabolic clinical biochemistry in humans.

To determine the effect of prospective changes in body weight on circulating EMC10 levels, we measured serum EMC10 levels in white human study participants before and 12 months after weight loss intervention. At 12 months after bariatric surgery BMI decreased by 30.1% ($P < 0.001$) from baseline and was associated with decreases in serum EMC10 levels by 57.9% ($P < 0.001$) and HOMA-IR by 77.4% ($P < 0.001$) (Fig. 2A–C, Supplementary Table 3). In human study participants who underwent combined hypocaloric diet and exercise for 12 months we observed a 32% reduction in serum EMC10 ($P < 0.01$) and a 46% reduction in HOMA-IR ($P < 0.001$), despite only a modest change in BMI (Fig. 2D–F, Supplementary Table 3). Moreover, reduction in circulating EMC10 positively correlated with changes of both BMI and HOMA-IR (Fig. 2G, H). In addition, HbA1c and serum triglycerides, ALT and AST all decreased 12 months after intervention in the weight-loss cohort (Supplementary Fig. 2E–L) and the changes in these parameters were all positively correlated with the changes in serum EMC10 levels (Fig. 2I–L).

In summary, our data demonstrate that circulating EMC10 is positively correlated with BMI and insulin resistance in humans, and weight loss interventions reduce serum EMC10. These findings implicate scEMC10 in the aetiology of obesity and its metabolic complications.

### Regulation of *scEmc10* expression in obese mice and people with obesity

Given that these data identify serum EMC10 as a potential biomarker of adiposity, we reasoned that its expression may be upregulated in obese adipose tissue. Therefore, we measured *scEMC10* expression by qPCR in subcutaneous adipose tissue from people with or without overweight and obesity. Surprisingly, *scEMC10* was actually downregulated in subcutaneous adipose tissue from volunteers with obesity compared to overweight and lean volunteers (Supplementary Fig. 3A, Supplementary Table 1). Therefore, the increase in circulating EMC10 is likely to be either derived from non-adipose tissue, or scEMC10 is regulated at the post-transcriptional level. Given that human tissue is limited, we analyzed *scEmc10* abundance across various metabolic tissues from lean and obese mice to help inform the source of circulating EMC10 in obesity. In lean mice, in addition to its high abundance in the iWAT, it is moderately expressed in the skeletal muscle, heart, BAT, and pancreatic islets (Supplementary Fig. 3B). Consistent with the effects of human obesity on *scEMC10* expression, *scEmc10* expression in iWAT (subcutaneous adipose tissue) was dramatically downregulated in obese mice after HFD treatment (Supplementary Fig. 3B). Downregulation of *scEmc10* in iWAT occurred as early as two weeks after the onset of HFD feeding (Supplementary Fig. 3C). These observations were confirmed in an independent rodent model of obesity – *ob/ob* mice (Supplementary Fig. 3D).

While many key metabolic tissues have unchanged *scEmc10* expression during obesity, we observed that *scEmc10* transcript was significantly upregulated in liver and pancreatic islets after HFD (Supplementary Fig. 3B). To identify the pathophysiological stimuli regulating *scEmc10* expression, we used animal models for insulin resistance and hepatic steatosis. We examined livers from acute insulin receptor knockdown (L-IR[KD]) mice, a lipodystrophic mouse model (IR[FKO])[24], and choline-deficient and methionine-restricted (CDA) diet-treated mice[25]. Our results showed significant upregulation of *scEmc10* in livers from IR[FKO] and CDA-treated, but not L-IR[KD], mice

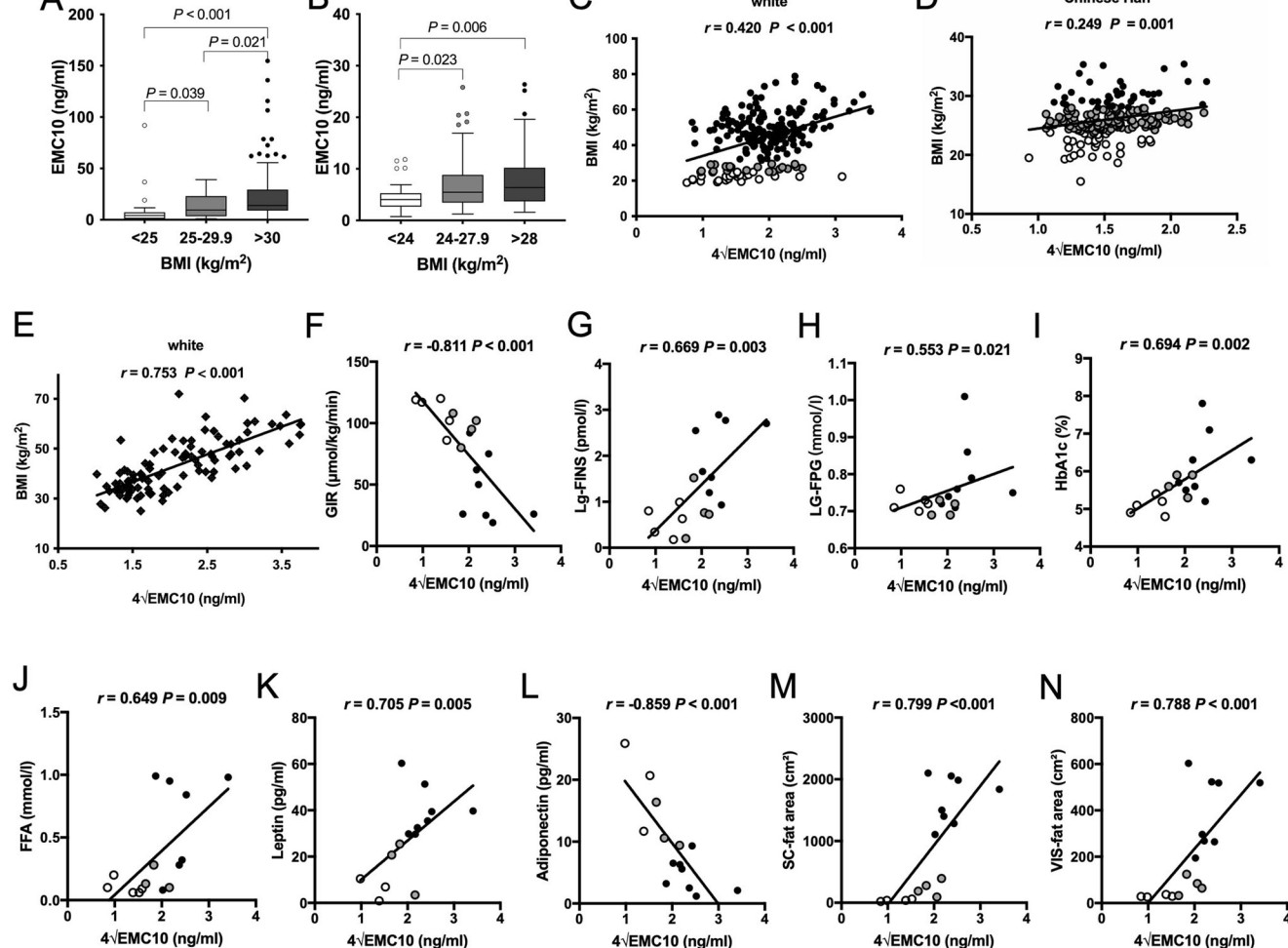

**Fig. 1 | Serum EMC10 levels in white and Chinese Han cohorts. A** Serum EMC10 levels in white human study participants with leanness ($n = 27$), overweight ($n = 20$) and obesity ($n = 160$). **B** Serum EMC10 levels in Chinese Han human study participants with leanness ($n = 32$), overweight ($n = 115$) and obesity ($n = 39$). Serum EMC10 levels are presented as box (median with interquartile range) and whisker (1.5x interquartile range) plots. Comparisons among 3 groups of participants were performed by one-way *ANOVA* with Fisher's *post hoc* test, and then multiple comparisons were performed using *LSD-t* test. A two-sided *P*-value of <0.05 was considered as significant. **C** Correlation of serum EMC10 levels with BMI in the white cohort shown in panel **A** ($n = 207$). **D** Correlation of serum EMC10 levels with BMI in the Chinese Han cohort shown in panel **B** ($n = 186$). **E** Correlation of serum EMC10 levels with BMI in a white weight-loss cohort ($n = 100$). The correlation analyses were performed using Pearson's bivariate correlation. **F−N** Correlation of serum EMC10 levels with GIR (glucose infusion rate), FINS (fasting plasma insulin), FPG (fasting plasma glucose), HbA1c, FFA (serum free fatty acid), serum leptin and adiponectin, and subcutaneous (SC) fat area and visceral (VIS) fat area, respectively, in white human study participants who underwent euglycemic-hyperinsulinemic clamp ($n = 17$). The correlation analyses were performed using Spearman's rank correlation analysis. Source data are provided in the Source Data file.

(Supplementary Fig. 3E). This suggests that *scEmc10* expression could potentially be regulated by hepatic steatosis, but not hepatic insulin resistance. The absence of insulin signaling regulation was further confirmed in insulin receptor KO (βIRKO) β-cells[26] (Supplementary Fig. 3F). In addition to pathological conditions, we investigated *scEmc10* expression after fasting and during acute refeeding to explore physiological processes that might regulate its expression. We observed that during fasting, *scEmc10* expression was dramatically downregulated in the liver compared to the fed state, but completely recovered after 2−4 h of refeeding (Supplementary Fig. 3G).

Besides pathophysiological regulation, we also examined *scEmc10* abundance across various metabolic tissues under physiological conditions such as cold exposure and thermoneutrality, we observed that cold exposure significantly downregulated *scEmc10* in BAT and iWAT but not in other tissues (Supplementary Fig. 3H). Taken together, our data clearly demonstrate that the expression of *scEmc10* is dysregulated in mouse models of metabolic disease.

## *Emc10* knockout mice are resistant to diet-induced obesity

We reasoned that the association of serum EMC10 with increasing BMI and fat mass could be a primary cause of increased adiposity or a consequence, partially mediating the metabolic complications of obesity. To explore the effect of EMC10 on energy balance and systemic metabolism, we generated a whole-body *Emc10* knockout (KO) mouse. Successful ablation was confirmed by the virtual absence of both *scEmc10* and *mEmc10* expression in all tissues examined (Supplementary Fig. 4A). The KO mice showed normal gross morphological features on a chow diet (CD) and exhibited body weights similar to wildtype (WT) controls, even up to 52 weeks of age (WT: 33.68 ± 1.64; KO: 33.77 ± 1.78; WT vs KO, $p = 0.97$). To further characterize the effects of *Emc10* knockout, we subjected mice to either a low-fat diet (LFD) (10% fat by kcal) or high-fat diet (HFD) (60% fat by kcal) and undertook metabolic phenotyping. The KO mice fed LFD for 12 weeks exhibited a trend towards reduced body weight (Supplementary Fig. 4B). This was mediated by a reduction in total adiposity as assessed

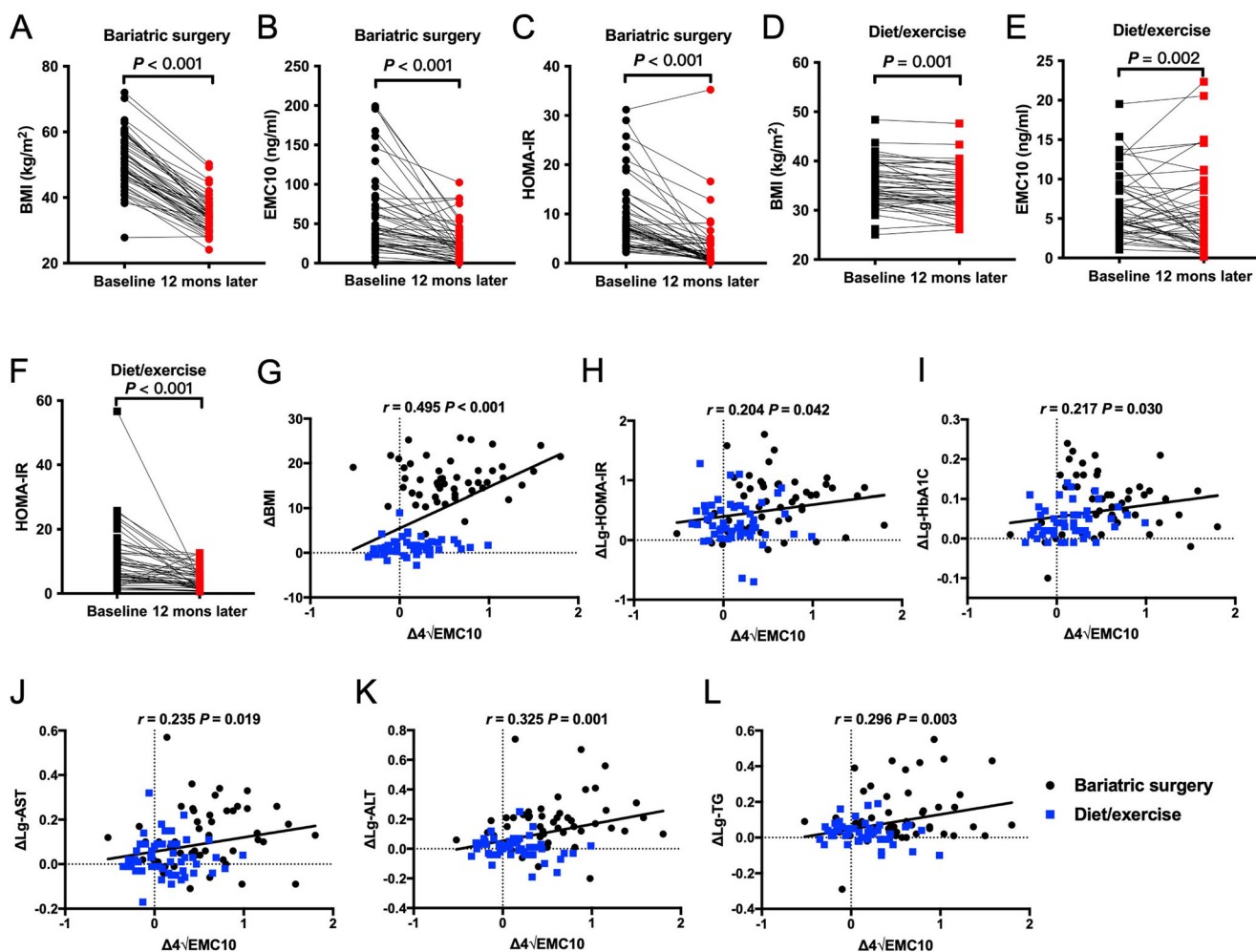

**Fig. 2 | Serum EMC10 levels in white weight-loss cohorts. A–C** BMI, serum EMC10 concentration, or HOMA-IR of white human study participants before and 12 months after bariatric surgery, respectively (*n* = 50), ***p* < 0.001. **D–F** BMI, serum EMC10 concentration, or HOMA-IR of white human study participants before and 12 months after diet/exercise weight-loss intervention, respectively (*n* = 50). Comparisons between before and after were performed using Student's paired *t* test. **G–L** Correlations between changes of serum EMC10 levels and changes of BMI, HOMA-IR, HbA1c, serum AST, ALT and TG, respectively, after weight-loss intervention by either bariatric surgery or diet/exercise in the white cohort (*n* = 100), performed using Pearson's bivariate correlation. Source data are provided in the Source Data file.

by percentage fat mass or adipose tissue weight while lean mass was non-significantly reduced (Supplementary Fig. 4C, D).

The effect of *Emc10* KO was accentuated in HFD-fed mice. *Emc10* KO mice on HFD were significantly leaner with attenuation in weight gain from as early as 2 weeks after initiation of HFD (Fig. 3A). The lower body weight of *Emc10* KO mice was largely accounted for by substantial reduction of both inguinal white adipose tissue (iWAT) and epididymal white adipose tissue (eWAT) weights (Fig. 3B), without an alteration in lean body mass (Fig. 3C, Supplementary Fig. 5A).

Our histological analyses revealed that the decrease in the KO fat mass on HFD is likely driven by a reduction in adipocyte size, as demonstrated by a significantly greater frequency of small adipocytes and lower frequency of mid-sized and large adipocytes in both the eWAT and iWAT (Supplementary Fig. 5B, C). This is also evidenced by the near-normal appearance of the KO brown adipocytes, compared to the enlarged, lipid-laden brown adipocytes harvested from the WT mice after HFD treatment (Supplementary Fig 5B). Similar, but more subtle, changes in the distribution of adipocyte size were also observed in the eWAT and brown adipose tissue (BAT) from the KO mice fed with LFD (Supplementary Fig. 4E, F). Consistent with a lean phenotype, *Emc10* KO mice fed a HFD

exhibited improved glucose tolerance and insulin sensitivity (Fig. 3D) and exhibited lower fasting glucose and insulin levels (Fig. 3E). Increased adipocyte size is positively correlated with leptin production[27]. Consistent with the larger adipocytes in the HFD-fed WT mice, we observed significantly higher leptin levels in the WT mice fed HFD, compared to the KO (Fig. 3F). Plasma adiponectin levels are decreased in obesity, insulin resistance, and type 2 diabetes[28]. In line with their leaner phenotype and improved metabolic profile, KO mice on HFD exhibited significantly higher serum adiponectin levels compared to WT controls (Fig. 3F). In addition to improved glucose metabolism, *Emc10* KO mice also exhibited significantly lower levels of fed serum triglyceride (TG), cholesterol (CHO), and non-esterified fatty acid (NEFA) levels (Fig. 3G). In contrast, in the LFD fed mice, we only observed trends towards increased plasma insulin and decreased leptin in the KO mice (Supplementary Fig. 4G).

Chronic exposure of mice to HFD causes hepatic steatosis[29]. Feeding a HFD, but not LFD, increased liver mass (Fig. 3B) and the number of large, lipid-containing vacuoles revealed by H&E staining (Fig. 3H, Supplementary Fig. 4H) in the WT livers, compared to KO livers. Accordingly, TG content of liver from KO mice was significantly lower following HFD (Fig. 3I). Additionally, adipose

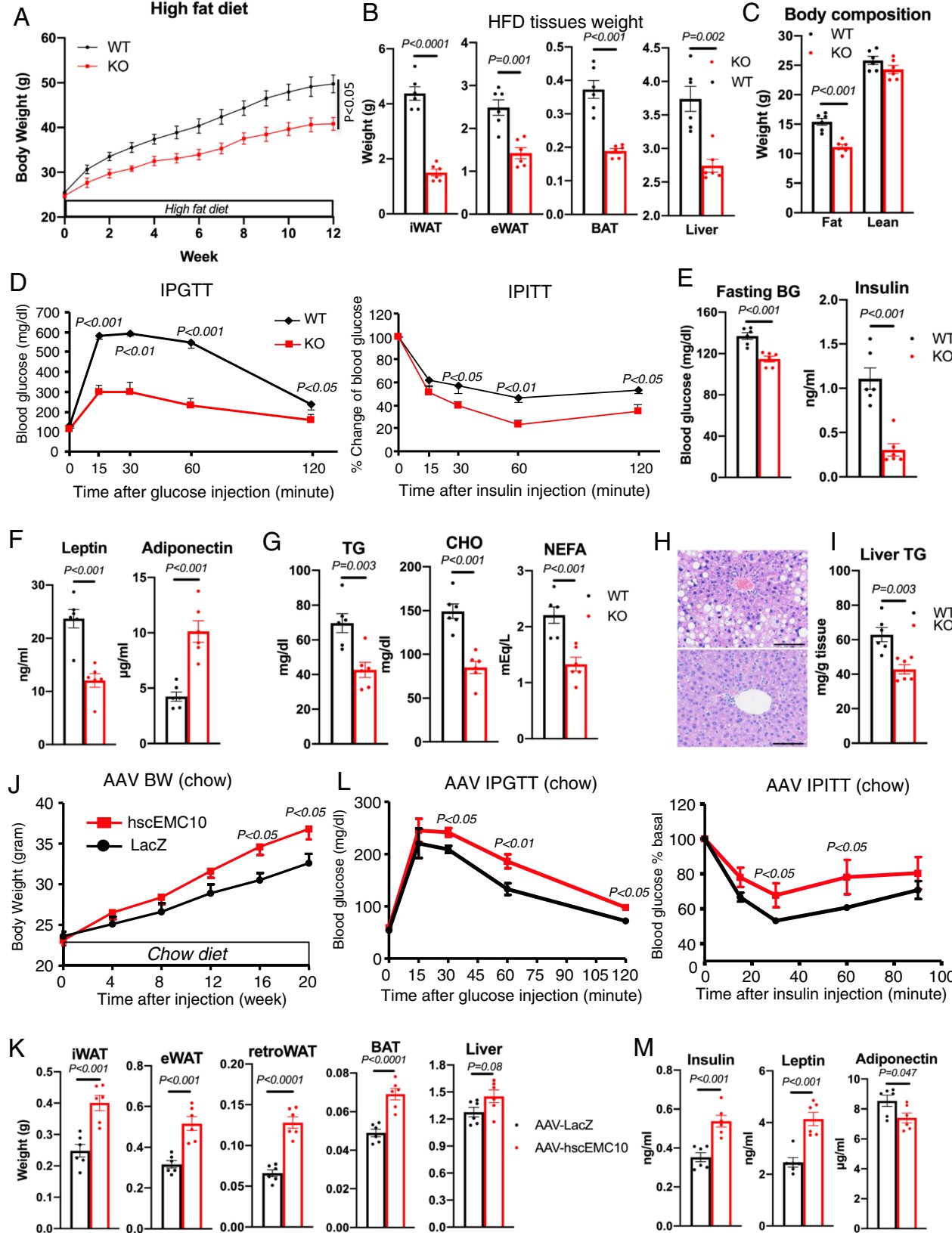

inflammation was improved in KO mice fed with HFD evidenced by significantly decreased gene expression of inflammation markers including *Mcp-1*, *Tnfa*, and *F4/80* in KO eWAT compared to WT (Supplementary Fig. 5D). In summary, *Emc10* KO protects mice from diet induced obesity.

## Upregulation of circulating EMC10 promotes obesity

To determine the metabolic consequences of increasing circulating EMC10, we performed intravenous injections of adeno-associated virus (AAV) encoding human *scEMC10* (*hscEMC10*) or LacZ to deliver full-length *scEMC10* or LacZ construct to the liver of 7-wk-old C57BL/6

**Fig. 3 | Effects of EMC10 ablation and scEMC10 overexpression on obesity and metabolic homeostasis. A** Body weights of male (WT, black circle), and KO (red square) on C57BL/6 background on HFD ($n$ = 6 & 7 per group). **B** Tissues (iWAT, eWAT, BAT, liver) weight from male WT (open), and KO (red) mice fed with 12-wks of HFD ($n$ = 6 per group). **C** Body composition of male WT (open) and KO (red) mice fed HFD by DEXA ($n$ = 6 per group). **D** Glucose tolerance (left) and insulin tolerance (right) in male WT (black diamond), and KO (red square) mice fed with 12-wks of HFD ($n$ = 8 & 7 per group). Plasma glucose and insulin (**E**); leptin and adiponectin (**F**); triglyceride (TG), cholesterol (CHO), and non-esterified free fatty acid (NEFA) (**G**), in male WT (open), and KO (red) mice fed with 12-wks of HFD in the fed or overnight fasted states ($n$ = 6 per group). **H** Representative images of H&E-stained sections of livers from male WT and KO mice fed with 12-wk of HFD. Scale bar, 100 um. **I** TG content of liver from male WT (open) and KO (red) mice fed with 12-

wks of HFD ($n$ = 6 per group). **J** Body weights of male C57BL/6 mice expressing LacZ control or *hscEMC10* via tail-vein AAV transduction after 20-wks of chow diet (CD) ($n$ = 6 per group). **K** Tissues (iWAT, eWAT, retroperitoneal (retroWAT), BAT, liver) weight of male C57BL/6 mice expressing LacZ control or *hscEMC10* via tail-vein AAV transduction after 20-wks of CD ($n$ = 6 per group). **L** Glucose tolerance (left) and insulin tolerance (right) of male C57BL/6 mice expressing LacZ control or *hscEMC10* via tail-vein AAV transduction after 20-wks of CD ($n$ = 6 per group). **M** Plasma insulin, leptin, and adiponectin of male C57BL/6 mice in the fed state expressing LacZ control or *hscEMC10* via tail-vein AAV transduction after 20-wks of CD ($n$ = 6 per group). All data are presented as mean +/− SEM. Statistical significance was assessed by two-sided Student's $t$ test and significant differences were indicated with $p$ values. Source data are provided in the Source Data file.

male mice. Animals were subjected to either chow diet (CD) or HFD feeding one week after the injection. This method generally results in robust expression of the protein in the liver and potential secretion into the plasma[30]. Ten days post-injection, we observed a ~10-fold increase in liver EMC10 protein abundance and ~5·6-fold increase in plasma EMC10 levels, as detected by western blotting with an anti-EMC10 antibody (Supplementary Fig. 6A). As suggested by our human data, we observed that mice over-expressing *hscEMC10* gained significantly more weight than LacZ-expressing controls, even on chow diet (Fig. 3J). The higher body weight of *hscEMC10* mice was largely contributed by increased WAT mass (Fig. 3K, Supplementary Fig. 6B). Consequently, the heavier *hscEMC10* mice were also more glucose intolerant, insulin resistant (Fig. 3L), and had significantly higher levels of plasma insulin and leptin, consistent with their obese phenotype (Fig. 3M, and Supplementary Fig. 6C).

Consistent with the chow diet data, we observed that increased circulating EMC10 promotes diet-induced obesity in mice as early as two weeks after introduction of HFD (Supplementary Fig. 6D). In line with the body weight phenotype, we observed that mice expressing *hscEMC10* are also more glucose intolerant and insulin resistant (Supplementary Fig. 6E). The increased body weight observed in *hscEMC10* mice is largely contributed by a significant increase in adipose tissue weight (Supplementary Fig. 6F). In addition to increased fat mass, *hscEMC10* over-expressors exhibited hyperinsulinemia, hyperleptinemia, hyperlipidemia, and showed significantly lower circulating adiponectin (Supplementary Fig. 6G). Taken together, our gain and loss of function experiments in mice demonstrate that scEMC10 is a regulator of energy balance and provides supportive evidence that the associations between serum EMC10 and BMI observed in humans with obesity are causal.

## EMC10 ablation promotes energy expenditure through the activation of adipose tissue thermogenesis

To determine how EMC10 alters energy balance in mice, we determined the food intake, gene expression of anorexigenic and orexigenic peptides in the hypothalamus, and the ability of the intestine to absorb fat in both WT and KO mice[31]. We found that the total food intake, hypothalamic anorexigenic, and orexigenic peptide gene expression, and intestinal fat absorption between the WT and KO mice were unchanged (Supplementary Fig. 7A–C). However, when the food intake data were normalized to body weight, the *Emc10* KO mice were hyperphagic relative to wildtype controls (Supplementary Fig. 7A).

Reduction in adipose tissue mass without significant alterations in energy intake suggested a potential increase in energy expenditure in the KO mice. Indeed, measurement of oxygen consumption (VO2) and carbon dioxide elimination (VCO2) rates over 48 h, including two cycles of light and dark phases, revealed significant increases in oxygen consumption and carbon dioxide production in the HFD-fed, but not LFD-fed, KO mice (Fig. 4A and Supplementary Fig. 8A), which could not be attributed to changes in activity (Supplementary Fig. 7D). Consistent with the higher VO2 and VCO2, *Emc10* KO mice also generated

more heat (Fig. 4B) and exhibited a ~0.8 °C higher basal core body temperature (Fig. 4C). To confirm, we reanalyzed the data with analysis of covariance (ANCOVA)[32]. Consistent with our prior analysis, the differences in energy expenditure remained significant even when lean mass was used as a covariate (Fig. 4D). The generally higher respiratory exchange ratio (RER = VCO2/VO2) in the HFD-fed *Emc10* KO mice indicated higher carbohydrate utilization than the more obese WT controls, despite both groups being fed HFD for 12 weeks (Fig. 4E). Consistent with the increased energy expenditure of the KO mice, we observed significantly decreased VO2, VCO2 and heat production in the mice over-expressing hscEMC10 compared to LacZ control (Supplementary Fig. 8B, C). Taken together, our data indicate that increased thermogenesis in the *Emc10* KO mice is the primary mechanism contributing to their resistance to diet-induced obesity.

Since increased adipose thermogenesis and metabolism can augment whole-body energy expenditure, we examined expression levels of markers of adipocyte differentiation, lipolysis, lipogenesis, and thermogenesis in both BAT and iWAT, as both types of fat play protective roles during obesity. We observed that ablation of *Emc10* robustly upregulated lipolytic, lipogenic, and thermogenic marker expression in BAT (Fig. 4F). This was also observed in the expression of lipolytic and thermogenic markers in iWAT harvested from HFD-fed *Emc10* KO mice, compared to WT controls (Fig. 4G). As HFD/obesity alone is known to have an impact on the expression of many of these markers, to further delineate which adipocyte function is the primary mechanism contributing to the obesity-resistant phenotype of the KO mice, we examined BAT and iWAT collected from LFD-fed mice. As expected, not all markers identified in the HFD-fed mice showed similar expression changes; however, the markers of thermogenesis did (Fig. 4H). To confirm changes in adipose tissue function, the oxygen consumption of BAT and iWAT harvested from the LFD-fed mice was measured using the Clark electrode. Consistent with the gene expression data, we observed that the ablation of *Emc10* significantly increased oxygen consumption in both the BAT and iWAT (Fig. 4I).

A key physiological difference in mice and humans is the proportion of energy devoted to maintaining body temperature when housed at ambient room temperature. This difference has key implications for the translation of mouse biology to humans, particularly with respect to thermogenesis and energy expenditure[33]. To demonstrate the robustness of our findings in this regard, we repeated our analysis of mouse body weight at thermoneutrality (30 °C). Both WT and KO mice were weaned and subsequently subjected to HFD treatment at thermoneutrality. Our data demonstrated that the KO mice were significantly leaner than the WT controls at thermoneutrality (Fig. 4J).

The differences in body weight at thermoneutrality and the effects of *Emc10* KO on adipose tissue phenotype suggest that EMC10 augments energy expenditure by increasing adipocyte thermogenic capacity and increasing non-shivering thermogenesis. To confirm the role of EMC10 in non-shivering thermogenesis, we measured oxygen consumption in WT and KO mice at thermoneutrality in response to a

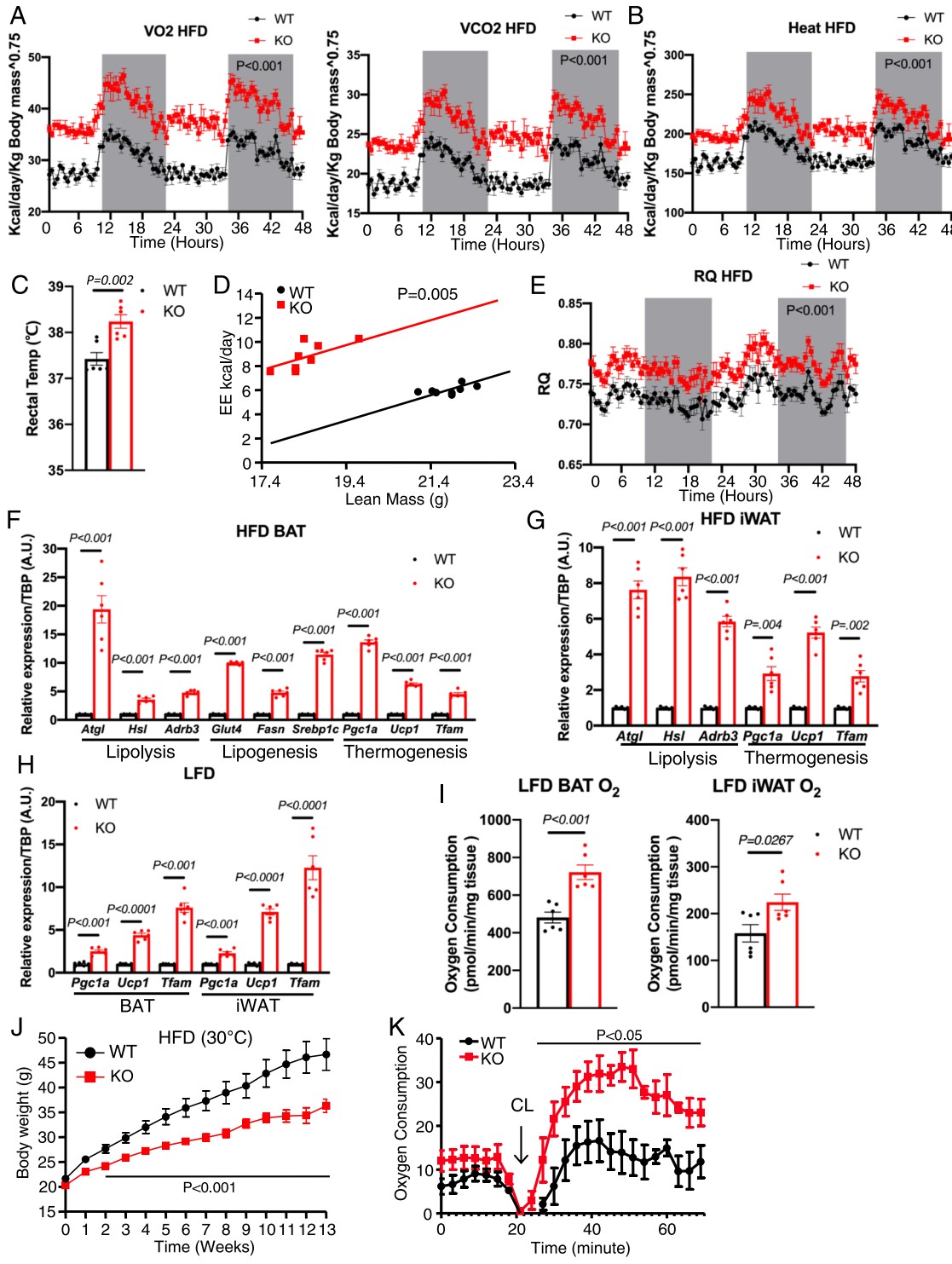

**Fig. 4 | EMC10 ablation promotes adipose tissue oxygen consumption and whole-body energy expenditure. A** Oxygen consumption (VO2) (left) and carbon dioxide production (VCO2) (right), **B** Heat production analyzed by indirect calorimetry for 48 h in male WT (black circle) or KO (red square) mice after fed with 12-wks of HFD ($n = 8$ per group). **C** Rectal temperature measured for male WT (black) and KO (red) mice fed HFD at room temperature ($n = 8$ per group). **D** Energy expenditure analyzed with ANCOVA using lean mass as covariate for WT (black circle) or KO (red square) mice ($n = 8$ per group). **E** Respiratory exchange ratio (RER) analyzed by indirect calorimetry in male WT (black circle) and KO (red square) mice fed with HFD ($n = 6$ per group). **F** *Atgl, Hsl, Adrb3, Glut4, Fasn, Srebp1c, Pgc1a, Ucp1*, and *Tfam* mRNA in BAT from male WT (balck) or KO (red) mice after fed with 12-wks of HFD ($n = 6$ per group). **G** *Atgl, Hsl, Adrb3, Pgc1a, Ucp1*, and *Tfam* mRNA in iWAT from male WT (balck) or KO (red) mice after fed with 12-wks of HFD ($n = 6$ per group). **H** *Pgc1a, Ucp1*, and *Tfam* mRNA in BAT and iWAT from male WT (black) or KO (red) mice after fed with 12-wks of LFD ($n = 6$ per group). **I** Oxygen consumption in BAT (left) and iWAT (right) from male WT (black) or KO (red) mice fed with 12-wks of LFD. ($n = 6$ per group). **J** Body weights of male WT (black), and KO (red square) on HFD at 30 °C ($n = 5$ & 6 per group). **K** Oxygen consumption of male WT (black circle) and KO (red square) before and after CL316, 243 (0.1 mg/kg) stimulation ($n = 5$ & 4 per group). All data are presented as mean +/− SEM. Student's *t* test was used for statistical analysis. Statistical significance was assessed by two-way ANOVA followed with Bonferroni's multiple comparisons test (**A**, **B** and **E**), two-sided Student's *t* test (**C**, **F**–**K**) and significant differences were indicated with *p* values. Source data are provided in the Source Data file.

β3 adrenergic agonist – CL316,234 – which is expected to activate thermogenesis only in brown adipose tissue where β3-adrenoreceptors are highly expressed[34]. Consistent with the obesity-resistant phenotype, the KO mice showed a significantly enhanced response to CL316,243 compared to WT control (Fig. 4K).

Our data demonstrate that loss of EMC10 enhances thermogenic capacity of adipocytes, increases energy expenditure and protects mice from diet-induced obesity.

## Ablation of EMC10 upregulates p38MAPK and CREB pathways

To investigate the underlying mechanisms promoting the increased oxygen consumption in the KO adipose tissues, we examined basal and β3 agonist-stimulated thermogenic marker expression in brown, inguinal, and epididymal adipocytes differentiated from primary SVF isolated from WT and KO mice. Our data showed that ablation of EMC10 significantly upregulated basal expression of *Ucp1* and the transcriptional regulator *Pgc1α* in all types of adipocytes examined (Fig. 5A–C). In addition, stimulation with the β3 agonist, CL316, 243, significantly increased *Ucp1* and *Pgc1α* expression in *Emc10* KO brown, inguinal, and epididymal adipocytes compared to controls (Fig. 5A–C).

To determine whether increased adipocyte thermogenesis is truly mediated by loss of EMC10, we treated the differentiated KO brown adipocytes with exogenous recombinant scEMC10. We observed that the basal *Ucp1* and *Pgc1α* upregulation in *Emc10* KO adipocytes were diminished by increasing doses of recombinant scEMC10 in the culture media (Fig. 5D). Consistent with the transcript data, western blotting also showed dramatically increased UCP1 protein in the differentiated KO adipocytes compared to WT and the UCP1 protein levels were reduced by increasing amount of exogenous recombinant, but not heat-inactivated scEMC10 (Fig. 5E). As *Emc10* KO adipocytes appeared to be highly responsive to β3 agonist stimulation, we first determined whether basal upregulation of *Ucp1* and *Pgc1α* is dependent on key downstream effectors of β3-adrenoceptors cAMP/PKA by using the PKA-specific inhibitor, H89. Our results showed that inhibiting the kinase activity of PKA completely abolished the upregulation of thermogenic markers in *Emc10* KO brown adipocytes (Fig. 5F).

To identify further downstream signaling targets, we examined PKA downstream signaling molecules that are known regulators of *Ucp1* and *Pgc1α*. Western blotting revealed robust upregulation of pCREB and p38MAPK in *Emc10* KO brown adipocyte lysates (Fig. 5G). To confirm that activation of the p38MAPK pathway is required for thermogenic marker induction, we treated KO brown adipocytes with the p38MAPK inhibitor, SB203580. Our results showed that the upregulation of *Ucp1* and *Pgc1α* gene expression was completely abolished by SB203580 treatment in the KO adipocytes (Fig. 5H). Similarly, treatment with the CREB inhibitor, HY-101120, also abolished upregulation of basal *Ucp1* and *Pgc1α* expression in the KO brown adipocytes (Fig. 5I).

To gain further mechanistic insight, we investigated whether scEMC10 is capable of directly regulating PKA activity. PKA is a holoenzyme composed of two regulatory subunits and two catalytic subunits. For both subunits, there are several isoforms. Among the isoforms of catalytic subunits, PKA catalytic alpha (PKA Cα) is the predominant isoform expressed in adipocytes[35]. Firstly we performed co-immunoprecipitation (Co-IP) of scEMC10 with PKA Cα in 293 T cells. We observed that using AKT1 as a control, exogenous scEMC10 directly binds to exogenous PKA Cα, but not exogenous AKT1 (Fig. 6A). Furthermore, we found that exogenous scEMC10 also directly binds to endogenous PKA Cα in 293 T cells (Fig. 6B). Next we performed an in vitro kinase assay to confirm the impact of scEMC10 on PKA activity. Our assays showed that recombinant, but not the heat-inactivated, scEMC10 inhibited CREB phosphorylation by PKA (Fig. 6C). Our data suggest that PKA signaling pathway is modulated by direct interaction with scEMC10. ScEMC10 is a secreted factor (Supplementary Fig. 1), while PKA Cα is an intracellular protein kinase

subunit. To determine whether extracellular scEMC10 could be transported into cell, we labeled recombinant scEMC10 with FITC and incubated with Hela cells. Our analysis clearly showed labeled scEMC10 within the cells (Fig. 6D). These data support our hypothesis that extracellular scEMC10 is transported into target cells via a currently unknown mechanism to mediate its metabolic effects.

Taken together, our results show that under basal conditions, scEMC10 suppresses the p38MAPK and CREB signaling pathways, leading to inhibition of adipocyte thermogenesis.

## Circulating EMC10 neutralization reduces body weight in obese mice

Our in vivo studies so far are limited in that our loss of function model cannot differentiate the effects of mEMC10 and scEMC10 and the effects in our overexpression paradigm may due to hepatic scEMC10 rather than circulating scEMC10. To reconcile these issues and investigate whether pharmacological inhibition of scEMC10 action could reduce body weight and improve diet-induced metabolic dysfunction, we generated and screened multiple clones of mouse anti-scEMC10 monoclonal antibodies for the ability to neutralize scEMC10 inhibitory effect on CREB phosphorylation. Using our in vitro assay, we identified two monoclonal antibodies (4C2 and 4B12-1) that could repeatedly block scEMC10-mediated CREB inhibition (Supplementary Fig. 9A).

To confirm the 4C2 antibody-neutralizing efficacy, we used an AAV-based scEMC10 over-expressor model since the methodology to detect and quantify circulating mouse scEMC10 is currently not readily available. Wildtype C57BL/6J mice were intravenously injected with AAV-*scEmc10*. Thirteen days after the AAV injection, mice were treated with either isotype-matching IgG control or 4C2 neutralizing antibody. Following our earlier experimental design, mice over-expressing scEMC10 were injected with a second dose of IgG or 4C2 antibodies 2 days later. Our results showed that even with elevated circulating scEMC10 levels, administration of 4C2 neutralizing antibody was still able to almost completely diminish its target in the blood (Supplementary Fig 9B).

With the newly generated monoclonal antibodies, we treated the C57BL/6J mice after 7 wk of HFD feeding. Obese B6 mice were injected with either isotype-matching IgG control, an anti-EMC10 monoclonal antibody (1F12) that could not neutralize scEMC10 effect in our in vitro assay or one of the two anti-EMC10 neutralizing antibodies twice a week. We observed that immediately following the immunological treatment, mice treated with both the neutralizing antibodies (4C2 and 4B12-1), but not the non-neutralizing 1F12 antibody or IgG control, demonstrated reduced body weights with stronger effects observed in mice treated with 4C2 antibody (Fig. 7A, Supplementary Fig 9C). To confirm the effect of 4C2 antibody on body weight loss, we performed an antibody-swapping experiment where after one-week treatment of 4C2 antibody or control IgG, the two antibodies swapped with each other to treat mice for 4 days followed by swapping back to original respective antibody for another 3-day treatment. We observed that as expected, one-week treatment of 4C2 antibody significantly decreased mouse body weights. After exchanging 4C2 for control IgG, the body weights of mice in the same group significantly increased, before going down after 4C2 antibody was re-instated (Fig. 7B). Similarly, in the control IgG group, the crossover to 4C2 antibody decreased mouse body weights, which then subsequently increased when 4C2 was withdrawn (Fig. 7B). These observations clearly demonstrate that scEMC10 inhibition promotes weight loss in mice.

Consistent with our mouse genetic studies, the lower body weights of the 4C2-treated mice were the result of decreased fat mass with a smaller contribution from changes in liver mass (Supplementary Fig. 9D, E). Our histological analyses showed that the decrease in the fat mass in 4C2-treated mice is driven by the reduction in adipocyte size in both the WAT and BAT (Supplementary Fig 9F). We observed *Emc10* KO protected from steatosis caused by HFD (Fig. 3H, I). Similarly, 4C2

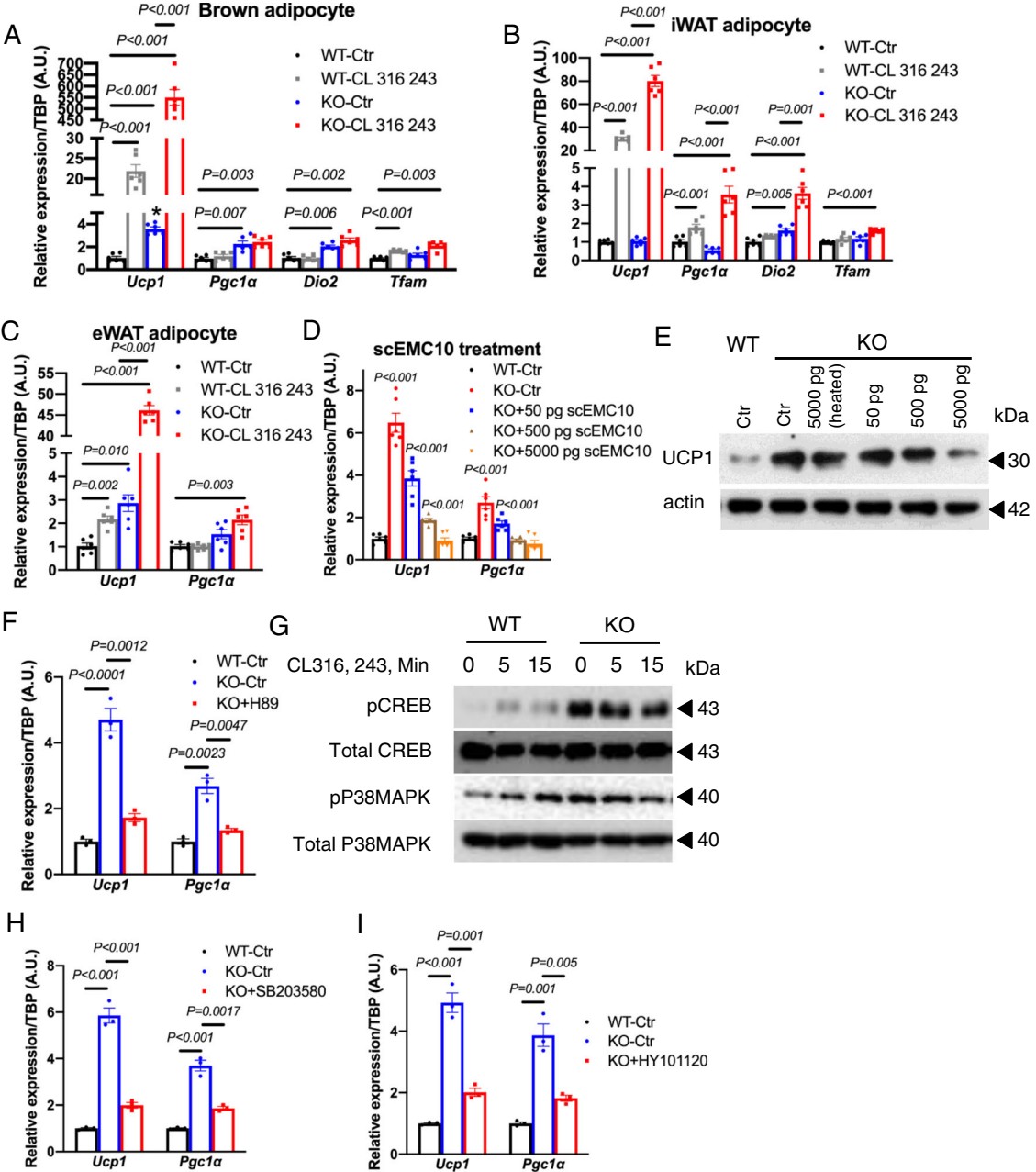

**Fig. 5 | EMC10 ablation activates adipocyte adaptive thermogenesis via PKA-mediated CREB and p38MAPK activities.** *Ucp1, Pgc1a, Dio2,* and *Tfam* mRNA in differentiated brown (**A**) or inguinal (**B**) primary adipocytes from male WT or KO mice treated with saline control or 0.1 µM CL316, 243 for 24 h (*n* = 6 per group). **C** *Ucp1* and *Pgc1a* mRNA in differentiated epididymal primary adipocytes from male WT or KO mice treated with saline control or 0.1 µM of CL316, 243 for 24 h (*n* = 6 per group). **D** *Ucp1* and *Pgc1a* mRNA in differentiated brown primary adipocytes from male WT or KO mice treated with control or recombinant human scEMC10 at indicated dose for 24 h (*n* = 6 per group). **E** Western blotting for UCP1 and β-actin in differentiated brown primary adipocytes from male WT or KO mice treated with control or recombinant scEMC10 at indicated dose for 24 h. **F** *Ucp1* and *Pgc1a* mRNA in differentiated brown primary adipocytes from male WT or KO mice

treated with control or PKA inhibitor, 1 µM of H89 for 24 h (*n* = 3 per group). **G** Western blotting for pCREB, total CREB, pP38MAPK and total p38MAPK proteins in differentiated brown primary adipocytes from male WT and KO mice after treated with saline (0 min) or CL316,243 (5, 15 min) (*n* = 3 per group). **H** *Ucp1* and *Pgc1a* mRNA in differentiated brown primary adipocytes from male WT or KO mice treated with control or p38MAPK inhibitor, 1 µM of SB203580 for 24 h (*n* = 3 per group). **I** *Ucp1* and *Pgc1a* mRNA in differentiated brown primary adipocytes from male WT or KO mice treated with control or CREB inhibitor, 0.5 µM of HY101120 for 24 h (*n* = 3 per group). All data are presented as mean +/− SEM. Statistical significance was assessed by two-sided Student's *t* test (**A–D**, **F**, **H** & **I**) and significant differences were indicated with *p* values. Source data are provided in the Source Data file.

antibody treatment for 3 weeks prevented mice from hepatic steatosis, as evidenced by histologically reduced ectopic lipid accumulation and decreased triglyceride content in liver (Fig. 7C, D). In addition to effects on body composition, scEMC10 neutralization also leads to significantly improved glucose tolerance and insulin sensitivity in obese mice 3 weeks of treatment (Fig. 7E, Supplementary Fig. 9G). We observed beneficial effects of scEMC10 neutralization on other

metabolic parameters, including significantly lower fasting blood glucose, ALT, and fed TG and NEFA, and higher adiponectin in obese mice treated with 4C2 antibody for 3 weeks, and significantly lower fed plasma TG and NEFA in 4B12-1 antibody-treated mice when compared with control antibodies (Fig. 7F–H, Supplementary Fig. 9H, I). In *Emc10* KO mice fed with HFD, we observed increased mRNA levels of thermogenic markers in BAT, which likely accounts for the enhanced

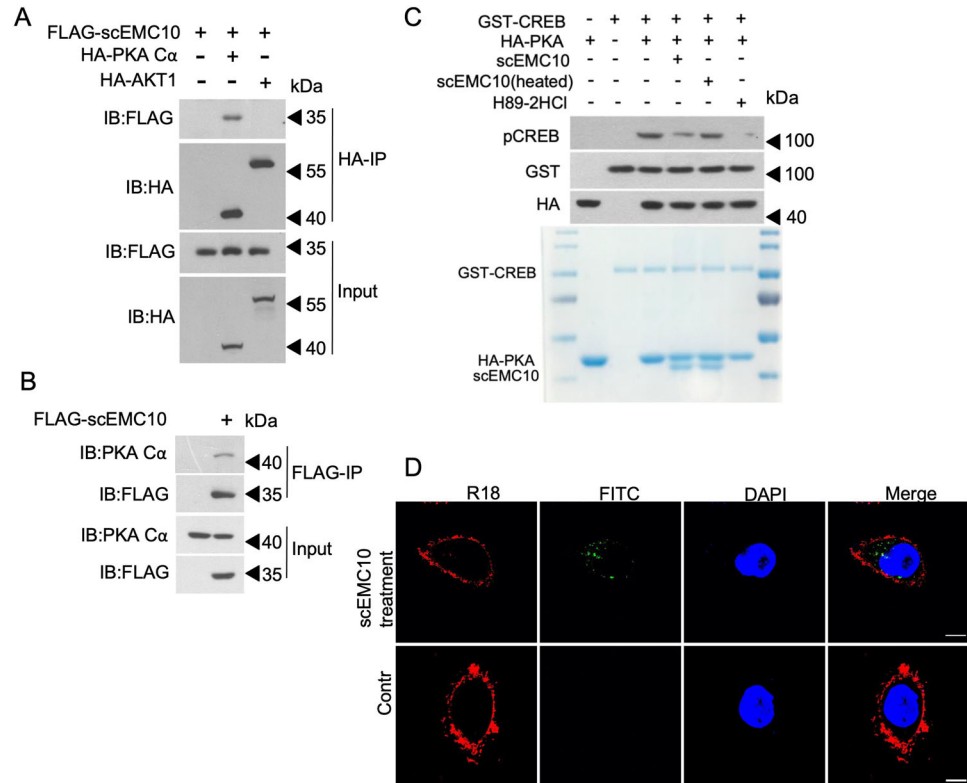

**Fig. 6 | Extracellular scEMC10 binds PKA Cα and inhibits its stimulatory action on CREB. A** Immunoprecipitation and western blotting of scEMC10, PKA Cα, and AKT1 in cell lysate prepared from 293T cells transfected with either FLAG-scEMC10 plasmid alone, or co-transfected with HA-PKA Cα or HA-AKT1 plasmid. **B** Immunoprecipitation and western blotting of scEMC10 and endogenous PKA Cα in cell lysate prepared from 293T cells transfected with FLAG-scEMC10 plasmid. **C** Western blotting for pCREB, total CREB and PKA in 293T cells after treatment with H89-2HCl (10 μM), recombinant scEMC10 protein (2 μg) or inactivation scEMC10 protein (2 μg) as indicated. **D** Immunofluorescence analysis of Hela cells treated with either scEMC10 labeled with FITC or control. Nuclei were stained blue with 4′,6-diamidino-2-phenylindole (DAPI), and cell membranes were stained red with Octadecyl Rhodamine B Chloride (R18). Scale bar, 5μm. Each experiment was repeated 3 times independently with similar results. Source data are provided in the Source Data file.

thermogenesis observed in these mice (Fig. 4B, F). Similar to the observations in the KO mice, mRNA levels of several thermogenic markers including *Ucp1*, *Pgc1α*, *Dio2*, *Cox8b*, and *Elvol3* were significantly increased in obese mice treated with 4C2 antibody (Fig. 7I). In agreement with the transcript data, protein levels of UCP1 and PGC1α markedly increased in BAT of these mice (Fig. 7J). Moreover, metabolic cage analysis revealed increased oxygen consumption, carbon dioxide elimination, and heat production in 4C2 antibody-treated obese mice, suggesting neutralization of circulating EMC10 promotes thermogenesis and energy expenditure (Fig. 7K).

Taken together, our proof-of-concept study demonstrates that immunological neutralization of circulating EMC10 promotes weight loss and improves obesity-induced metabolic dysfunction in obese B6 mice.

## Discussion

The alarming rise in the prevalence of obesity in recent decades has invigorated interest in the mechanisms of obesity and its complications. Dysregulation in anorectic signals and their sensing results in chronic overnutrition, accrual of fat mass and results in obesity. In line with this model, approved drugs for the treatment of obesity principally modulate central sensing of appetite[36]. However, the burden of obesity clearly mandates the development of a complimentary therapeutic strategies. To this end, modulation of adipocyte thermogenic capacity has garnered intense interest in the past decade but has yet to be successfully leveraged as a therapeutic strategy[37–40]. Though multiple endocrine modulators of thermogenic capacity have been described, their therapeutic utility is uncertain. For example, although FGF21 promotes browning of white fat and modulates thermogenic

activity independently via central actions[11,41,42], it is paradoxically elevated in obesity raising the suggestion that obesity may be an FGF21-resistant state[43]. The identification of a circulating modulator of thermogenesis that is dysregulated in a manner consistent with a role in driving weight gain will identify promising targets for the treatment of obesity.

In this study, we identified an evolutionarily conserved, circulating modulator of thermogenesis, scEMC10. Our current data suggest a model whereby scEMC10 is an endocrine factor that regulates adipocyte thermogenic capacity via inhibition of PKA and modulation of key transcriptional regulators of adipocyte thermogenesis. Consistent with a role in the pathogenesis of human obesity, circulating EMC10 exhibits striking positive correlations with indices of adiposity in humans.

Using three orthogonal mouse models, we clearly demonstrate that modulation of scEMC10 changes body weight in mice via changes in expenditure. Gene expression analyses in mouse adipose tissue and our ex vivo studies demonstrate that adipose tissue from these mice exhibits enhanced thermogenic capacity. These effects are quick in onset and seem rapidly reversible – as evidenced by the effects of recombinant scEMC10 in isolated adipocytes and weight regain when scEMC10 neutralizing antibody is withdrawn from obese mice. The kinetics suggest scEMC10 predominantly regulates energy expenditure by activating thermogenesis in mouse adipocytes rather than driving adipocyte reprogramming to more thermogenic lineages.

From a mechanistic perspective, we have demonstrated that extracellular scEMC10 can be transported into cells where it binds to the catalytic subunit of PKA and inhibits its activity in vitro and this

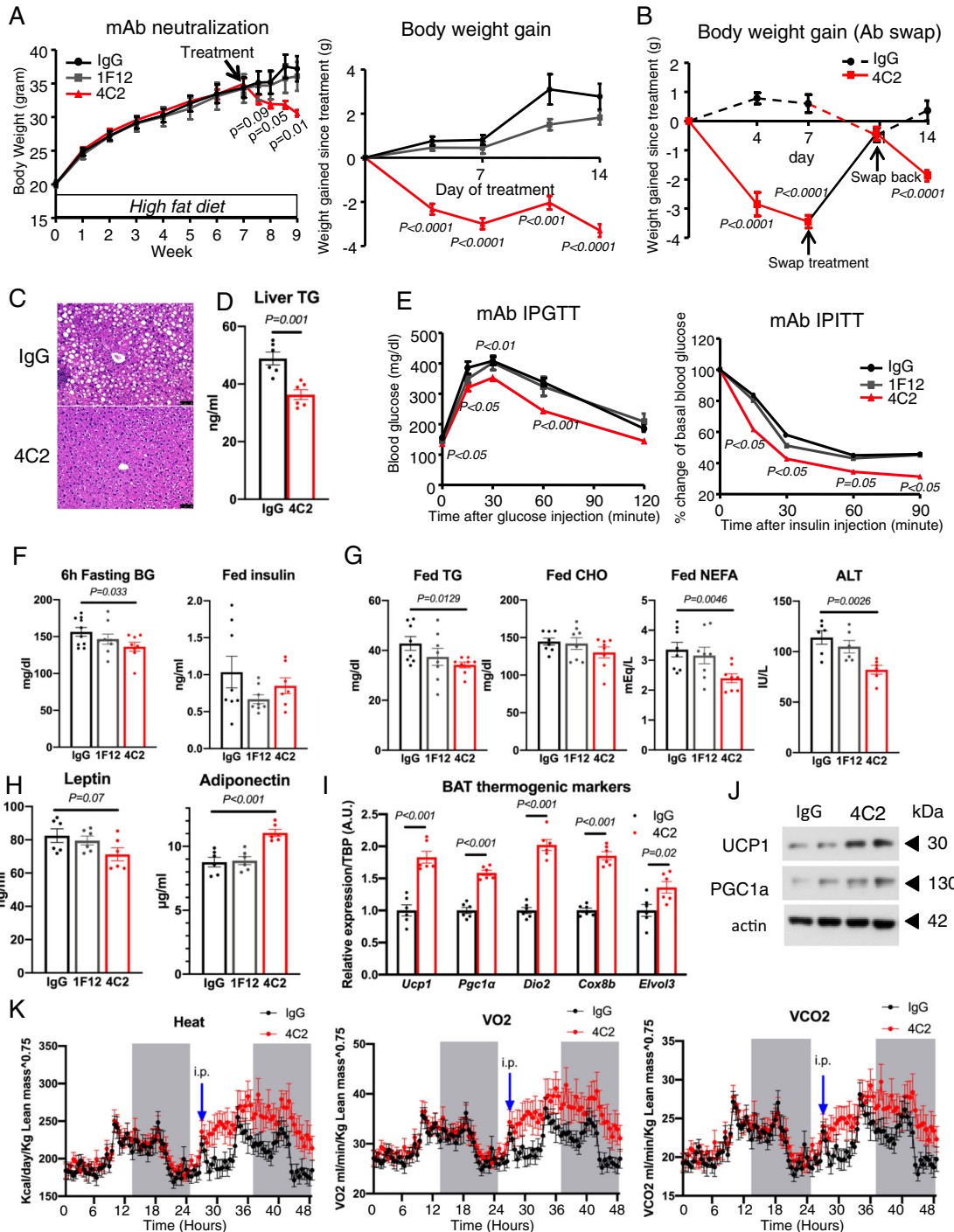

**Fig. 7 | Effects of scEMC10 neutralization on diet-induced obesity and metabolic homeostasis. A** Body weight (left) and body weight gain (right) of C57BL/6J male mice fed with HFD before or after IP injected with 3 mg/kg BW antibody as indicated twice a week. (IgG, black circle. mAb-1F12, gray square. mAb-4C2, red triangle) (*n* = 9 per group). **B** Body weight gain of HFD male mice IP injected with 3 mg/kg BW 4C2 antibody or control IgG twice a week. 7 days later, the two antibodies swapped with each other, and then 4 days later, swapped back followed by 3 other days of original treatment (IgG, dashed/black. mAb-4C2, solid/red) (*n* = 5 per group). **C** Representative images of H&E-stained sections of liver from male mice treated with IgG or 4C2 antibodies. Scale bar, 50 um. **D** TG content of liver from male mice treated with IgG (black) or 4C2 (red) antibodies for 3 weeks (*n* = 6 per group). **E** Glucose tolerance (left) and insulin tolerance (right) of male mice treated with IgG (black circle), 1F12 (gray square) or 4C2 (red triangle) antibodies (*n* = 9 per

group) for 3 weeks. Plasma glucose after 6 h fasting and fed insulin (**F**); Fed plasma triglyceride (TG), cholesterol (CHO), non-esterified fatty acid (NEFA), and ALT (**G**); plasma leptin and adiponectin (**H**) in male mice treated with IgG (black), 1F12 (gray) or 4C2 (red) antibodies for 3 weeks (*n* = 8 per group). **I** *Ucp1, Pgc1a, Dio2, Cox8b,* and *Elvol3* mRNA in BAT from male mice treated with IgG (black) or 4C2 (red) antibodies (*n* = 6 per group). **J** Western blotting for UCP1, PGC1a and actin in BAT from male mice treated with IgG or 4C2 antibodies. **K** Oxygen consumption (VO2), carbon dioxide production (VCO2), and heat production analyzed by indirect calorimetry for 48 h in male mice IP injected with IgG (black circle) or 4C2 (red circle) antibodies (*n* = 8 per group). All data are presented as mean +/− SEM. Statistical significance was assessed by two-sided Student's *t* test (**A, B, D–I**) and significant differences were indicated with *p* values. Source data are provided in the Source Data file.

reduces stimulatory phosphorylation of CREB and therefore inhibits CREB target gene expression. The classical modes whereby secreted proteins or circulating factors operate are to bind their cell surface receptors and trigger downstream signaling cascades. Over the past decades, a large body of evidence suggests many extracellular protein hormones, growth factors, and cytokines enter cytosol and nucleus and act intracellularly, among which fibroblast growth factors and interferon γ are well-established molecules that operate in this manner[44]. In this study, we also presented evidence that scEMC10 can enter cytosol and inhibit PKA activity. It is well recognized that activation of the PKA signaling pathway plays a direct regulatory role in the modulation of adaptive thermogenesis[40,45], thus our proposed mechanism of action is coherent with known adipocyte biology.

The motivation of our mouse studies was to characterize the role of scEMC10 in the pathophysiological context of obesity, given the striking correlations between adiposity and circulating EMC10 observed in our observational studies in humans. As such we have not fully explored the physiological function of scEMC10 in mice. However, it is of note that is acutely downregulated in adipose tissue by cold exposure and can modulate thermogenesis – findings that are consistent with scEMC10 being a physiological modulator of thermogenic tone in response to cold stress. Confirmation of this hypotheses with more detailed study and understanding how scEMC10 expression may become uncoupled from thermal stimuli in obesity are potential areas for further investigation.

The cAMP/PKA/CREB axis has broad regulatory actions in a range of tissues and, therefore, it is likely that scEMC10 exerts regulatory actions beyond adipose tissue that are independent of its effects on thermogenesis or energy expenditure. Elucidating these functions is beyond the scope of this study, but we expect this is fertile ground for further discovery and will have important implications for any development of scEMC10 as an obesity therapeutic. However, it is worth noting that PKA dependent signaling modulates glucoregulatory processes in liver whereby the net effect is to elevate blood glucose[46]. This is not consistent with the beneficial effects we see on glucose homeostasis when scEMC10 is inhibited in mice. It may simply be that the beneficial effects on adiposity exceed any acute glucose raising effects or glucose-lowering actions of PKA in other tissues outweigh the effects in liver. These explanations notwithstanding - these findings raise the intriguing possibility of tissue specific actions of scEMC10 mediated via selective uptake, differential expression of unidentified inhibitors of scEMC10 or varying sensitivity of different PKA isoforms to scEMC10.

Our studies in humans reveal a striking correlation between indices of adiposity and circulating EMC10 in both European and Chinese Han cohorts – thus scEMC10 is an alternative biomarker of adipose tissue mass. While we initially reasoned that adipose tissue was the probable source of scEMC10 upregulation in obesity, this hypothesis is not consistent with the fact that scEMC10 is actually decreased in the subcutaneous adipose tissue of obese mice and humans with obesity. Given the breadth of expression of scEMC10 it is difficult to ascertain the relative contribution of each tissue to circulating EMC10 and it is possible that different tissues contribute variable amounts depending on the nature of the stimuli. Development of assays to measure circulating mouse EMC10 with adequate sensitivity and tissue specific *Emc10* knockout mice will likely be pre-requisite tools to answer this question in future studies.

In addition to correlations with fat mass we also demonstrated remarkably consistent effects of weight loss interventions on serum EMC10 in humans and moderate-strong correlations of changes in circulating EMC10 concentrations with changes in BMI and various indices of metabolic health. Unfortunately, our studies do not have the temporal resolution to determine if changes in scECM10 precede or follow changes in fat mass, but our mouse studies raise the possibility that changes in scEMC10 could drive changes in fat mass in response to weight loss interventions. Further work understanding how metabolic surgery, diet and exercise regulate scEMC10 and observational studies of humans with high frequency serum sampling are warranted to answer this important question.

The therapeutic potential of modulation of adipocyte thermogenesis remains to be realized. scEMC10 is an attractive therapeutic target in this regard as it is upregulated in obesity and inhibits thermogenesis. Using antibody neutralization of scEMC10 we demonstrate that directly inhibiting EMC10 in the circulation can activate a thermogenic gene program in adipocytes, increase energy expenditure and induce weight loss. The robustness of this observation was demonstrated in a cross-over study whereby withdrawal of the antibody in exchange for placebo resulted in weight regain that was quickly abated when neutralizing antibody was re-introduced. These experiments provide an important a proof of concept that therapeutic inhibition of scEMC10 is effective and feasible

While our observational data in humans are consistent with an obesogenic effect of scEMC10 and our mouse data suggest a causal role for scEMC10 in this context, definitive confirmation of this hypothesis will require the conduct of interventional studies of scEMC10 administration or inhibition in humans. As outlined here, antibody mediated neutralization is effective and feasible in mice, but enhanced understanding of the regulation of scEMC10 at a transcriptional and post-transcriptional level may reveal other pharmacological modulators of circulating EMC10 that can be employed to test this hypothesis. Other translational applications of our work that should be considered are that scEMC10 could be used as a treatment for cachexia in cancer and other conditions. Work is ongoing to characterize circulating EMC10 concentrations in patients with cancer with and without cachexia.

In summary, we have identified scEMC10 as a circulating modulator of energy balance. Our work has direct implications for our understanding of human energy metabolism and identifies scEMC10 as an alternative therapeutic target for the treatment of obesity.

## Methods

### Human study participants for EMC10 serum concentrations

**Group 1.** We included 240 white individuals with either leanness (BMI < 25 kg/m², $n = 30$), overweight (BMI 25–30 kg/m², $n = 22$) or obesity (BMI > 30 kg/m², $n = 188$) who underwent abdominal surgery for cholecystectomy, weight reduction surgery, abdominal injuries or explorative laparotomy into our cross-sectional study of EMC10 serum levels and *scEMC10* mRNA expression in visceral and subcutaneous adipose tissues (Supplementary Table 1). All human study participants had a stable weight, defined as the absence of fluctuations of >2% of body weight for at least 3 months before surgery. We defined the following exclusion criteria: (1) Thyroid dysfunction, 2) alcohol or drug abuse, (3) pregnancy, (4) treatment with thiazolidinediones.

This study was approved by the ethics committee of the University of Leipzig (approval numbers: 159-12-21052012 and 017-12-23012012) following the principles of the Declaration of Helsinki. All human study participants gave written informed consent before taking part in the study.

**Group 2.** A total of 186 Chinese human study participants with either leanness (BMI < 24 kg/m², $n = 32$), overweight (BMI 24–28 kg/m², $n = 115$) or obesity (BMI > 28 kg/m², $n = 39$) who were recruited for diabetes screening were also enrolled in the cross-sectional study (Supplementary Table 2). Chinese human study participants with the following conditions were excluded: histories of diabetes, acute or chronic inflammatory disease, heart, liver or renal failure, cancer, or active use of oral hypotensive, hypolipidemic, anti-diabetic

medications. Serum EMC10 levels were investigated in the human study participants of this cohort.

This study was approved by the human research ethics committee of Huashan hospital following the principles of the Declaration of Helsinki. All human study participants gave written informed consent before taking part in the study.

**Group 3.** In two observational studies, we measured circulating EMC10 before and 12 months after a combined exercise and calorie restricted diet study ($n = 50$), before and 12 months after bariatric surgery ($n = 50$) (Supplementary Table 3). In this cohort, lifestyle and surgical interventions were part of the patients' clinical care. We defined the following exclusion criteria: (1) Thyroid dysfunction, (2) alcohol or drug abuse, (3) pregnancy, (4) treatment with thiazolidinediones.

This study was approved by the ethics committee of the University of Leipzig (approval numbers: 159-12-21052012 and 017-12-23012012) following the principles of the Declaration of Helsinki. All human study participants gave written informed consent before taking part in the study.

### Generation of mouse monoclonal antibodies against human scEMC10

Mouse monoclonal antibodies against human scEMC10 were generated using hybridoma methodologies (Phrenzer Biotechnology, Shanghai, China). Briefly, human *scEMC10* was expressed in 293 cells. Recombinant scEMC10 was purified from the supernatant medium and 100 μg was used to immunize each female BALB/c mouse at the age of 6–8 weeks every 2–3 weeks for 4 times. Lymphocytes were subsequently isolated from spleens of the immunized mice and fused with Sp2/0-Ag14 cells to form hybridoma cells. The supernatants of the hybridoma cells were used to react with scEMC10 by ELISA for screening out positive hybridoma cells. scEMC10-specific cells were sorted and then underwent further subcloning. scEMC10-specific hybridoma cells were injected intraperitoneally into BALB/c mice. Ascites was collected in which antibodies were subsequently purified using antigen affinity chromatography. In total, eight mouse monoclonal antibodies against human scEMC10 were validated by ELISA, among which mAb 6B9 and 1F12 were selected as coating and detecting antibody for the sandwich CLIA (chemiluminescent immunoassay) to detect scEMC10 in human serum (Phrenzer Biotechnology, Shanghai, China), respectively.

### Measurement of scEMC10 in human serum using double sandwich CLIA

The CLIA kits for detecting scEMC10 in human serum were obtained from Phrenzer Biotechnology. Briefly, 96-well immunoplates were coated overnight at 2–8 °C with mouse anti-scEMC10 mAb 6B9 at 500 ng/well. Each well was then blocked with blocking buffer (PBS, 0.5% bovine serum albumin, 10% sucrose) at 37 °C for 2 h. After aspirating each well and drying at room temperature for about 24 h, the immunoplates were ready for use. For performing a CLIA experiment, firstly, add 50 μl of human serum sample or scEMC10 standard to each well. Dispense 50 μl (dilution at 1:50,000) of scEMC10 mAb 1F12 HRP-conjugate into each well. Seal the immunoplates with acetate plate sealers. Mix all the wells gently with a shaker at 300–400 rpm for 15 s. Incubate the immunoplates overnight at 2–8 °C. Aspirate each well and wash with 350 μl of 1X washing buffer for 4 times. Add 100 μl of prepared substrate solution (50 μl substrate A and 50 μl substrate B) into each well. Mix all the wells and then incubate for 5–8 min at room temperature in a dark environment avoiding any sunlight. Determine relative luminescence units (RLU) of each well using a chemiluminescent microplate reader. For calibration of each sandwich CLIA, standards of 0, 0.3, 1.5, 7.5, 30, and 120 ng/mL recombinant scEMC10 protein were run in parallel with each testing plate. The CLIA system had an intra- and inter-assay coefficient of variation at 3.3–13.8% and 12–16.3%, respectively.

### Animals

Mice were housed in environmentally controlled conditions (temperature: 20–24 °C; humidity: 45–65%) with a 12-h light/dark cycle and had free access to standard rodent pellet food and water. The animal protocols #15-026 &18-010 were approved by the Institutional Animal Care and Use Committee (IACUC) of University of Illinois at Chicago. Humane endpoints were not established for this study, however, in addition to the daily routine general health inspection performed by the facility staffs, body weights were monitored weekly by the investigator lab members. Animal lost more than 20% of body weight will be excluded from the study and sacrificed. Euthanasia was done first exposing animals to CO2 and then followed by cervical dislocation. Animal care was given in accordance with institutional guidelines. Male C57BL/6J and *ob/ob* mice were obtained from the Jackson Laboratory (USA). *Emc10* transgenic animals used in this study are on a C57BL/6 background.

### Generation of *Emc10* knockout mouse model

The gene-targeting strategy was established on the basis of the mouse genomic DNA sequence (ENSMUSG00000008140). The target vector was achieved by ET cloning[47]. Two loxp elements were inserted to flank exon 2 of *Emc10* gene and a neomycin-resistant element was inserted between intron 2 and intron 3 for obtaining the targeting vector by homologous recombination in bacteria. The linearized vectors were electroporated into embryonic stem (ES) cells derived from 129 Sv/Ev mice (SCR012, Chemicon Ltd.). Neomycin-resistant ES cell colonies were isolated and expanded. Targeted ES cells were microinjected into the blastocysts of C57BL/6J female mice and transferred into the uteri of pseudo-pregnant mothers. Heterozygous mice were generated from mating of chimeric and C57BL/6J mice. The heterozygous mice were crossed with Flp recombinant mice (003800, Jackson lab) for deleting the neomycin-resistant elements. Then the Neo-deleted mice were crossed with EIIa-Cre mice (003314, Jackson lab) for deleting the flox field and obtaining the conventional knockout (KO) mice. The heterozygous mice have been backcrossed into C57BL/6j background for more than 10 generations.

### Overexpression of scEMC10 in vivo

To overexpress circulating scEMC10, we tail vein injected adeno-associated virus encoding human *scEMC10* or LacZ ($1.5 \times 10^{11}$ vg, Viral Core, Boston Children Hospital) to the liver of 7-wk-old C57BL/6 male mice. To determine the efficacy of scEMC10 neutralizing antibody, we tail vein injected AAV encoding mouse *scEmc10* ($2.5 \times 10^{11}$ vg) to the liver of 6-wk-old C57BL/6J mice. 13 days after injection, serum was collected to determine the expression of scEMC10 before first does of scEMC10 neutralizing antibody or IgG was injected (9 mg/kg) intraperitoneally. Second dose of antibody was injected on day 15 after the AAV administration. Serum was collected daily between day 13–17 to determine the circulating concentration of mouse EMC10.

### Body weight study

For diet-induced obesity, all mice were fed a chow diet (17% fat, 25% protein and 58% carbohydrate by kcal; #7012, Envigo) until 6 weeks of age. Subsequently, mice were assigned randomly to either a low-fat (10% fat, 20% protein, and 70% carbohydrate by kcal; D12450J, Research Diets) or a high-fat diet (60% fat, 20% protein, and 20% carbohydrate by kcal; D12492, Research Diets) until the end of the experimental protocol. Body weight was measured weekly until 18 weeks of age. For the thermoneutrality experiments, mice were adapted to 30 °C in an environmental chamber for at least 2 weeks before subjected to experimentation.

## Physiological studies and Histological Analyses

Blood glucose was monitored with an automated glucose monitor (Glucometer Elite, Bayer). Glucose tolerance tests (GTT) and insulin tolerance tests (ITT) were performed 16 h after fasting. For GTT, mice were injected intraperitoneally with glucose (2 g/kg body weight). Whole venous blood was obtained from the tail vein at 0, 15, 30, 60, and 120 min after the injection. For ITT, mice were injected intraperitoneally with human regular insulin (1 U/kg body weight). Whole venous blood was obtained from the tail vein at 0, 15, 30, 60, and 90 min after the injection[48]. Mice were anesthetized, and tissues were rapidly dissected, weighed and processed for immunohistochemistry. Hematoxylin and eosin (H&E) staining was performed on paraffin sections of liver and adipose tissues including epididymal, subcutaneous, and brown adipose tissues.

## Metabolic parameters

Plasma insulin was measured with an ELISA kit (Millipore). NEFA, TG, and cholesterol concentration in serum were measured with NEFA-C and Triglyceride E tests (Wako), respectively. Serum adiponectin and leptin levels were measured with ELISA kits from R&D Systems.

## Tissue triglyceride analysis

Lipids from tissues were extracted with Folch solution consisting of a mixture of 2:1 (vol/vol) chloroform/ methanol[49]. Lipids were solubilized in 1% Triton X-100 before evaporation under nitrogen gas. Triglyceride content was determined using the Triglyceride Determination Kit (Sigma).

## Adipocyte size determination

Adipocyte cross-sectional area from hematoxylin- and eosin-stained adipose tissue images (150–200 adipocytes/mouse, 4 mice/genotype) was calculated using ImageJ software.

## Food intake, energy expenditure, physical activity, and body composition

Food intake, physical activity, oxygen consumption (VO2), carbon dioxide (VCO2) and heat production were measured using the Comprehensive Laboratory Animal Monitoring System (CLAMS; Columbus Instruments). The respiratory exchange ratio ($V_{CO2}/V_{O2}$) was calculated from the gas exchange data and all data were normalized to lean body mass. Body composition (fat and lean mass) was assessed by the Dual-Energy X-Ray Absorptiometry (DEXA).

## Adipose tissue oxygen consumption

Adipose tissue oxygen consumption was performed using a Clark electrode (Strathkelvin Instruments). Freshly isolated tissues were isolated from WT or KO mice. Tissues were minced and placed in respiration buffer (DPBS containing 2% BSA, 0.45% glucose, and 0.012% pyruvate). For each depot, readings were taken with three separate pieces of tissue of equivalent size. O2 consumption was normalized to tissue weight.

## Thermoneutrality oxygen consumption measurement

7 - 8-week old *Emc10* KO&WT mice fed with chow diet were housed at 30 °C for 12 weeks. After acclimated to metabolic cage for 2 days, 0.1 mg/kg CL316,243 was injected intraperitoneally. Oxygen consumption as recorded before and after the injection.

## Primary stromal-vascular fraction isolation and differentiation

White adipose tissues: gWAT or iWAT were dissected from 7-8-wk-old WT and *Emc10* KO mice, minced, and digested with collagenase Type 1 (Worthington) (1 mg/ml in KRBA containing 125 mM NaCl, 4.74 mM KCl, 1 mM CaCl2, 1.2 mM KH2PO4, 1.2 mM MgSO4, 5 mM NaHCO3, 25 mM Hepes (pH 7.4), 3.5% BSA + 5.5 mM glucose) for 30–45 min with shaking at 37 °C[50], then centrifuged at $450 \times g$ for 5 min. The top

adipocytes were gently transferred as mature adipocytes and the SVF pellet was washed twice with KRBA and once with DMEM complete medium. Then, the pellet was resuspended in 5 ml of DMEM complete medium and filtered over 40 μm filter adaptor. Filtered SVF was plated onto rat tail collagen-I (Corning)-coated dish. For differentiation, confluent primary preadipocytes were differentiated with 50 nM insulin, 100 nM T3, 0.125 mM Indomethacin, 0.5 mM IBMX and 5 μM dexamethasone in DMEM/F12 media supplemented with 10% FBS for 2 days, followed by 4 days in medium supplemented with 50 nM insulin and 1 nM T3 with media change in between.

Brown adipose tissue: BAT were dissected from 7-8-wk-old WT and *Emc10* KO mice, minced, and digested with collagenase Type II (Worthington) (2 mg/ml in KRBA containing 125 mM NaCl, 4.74 mM KCl, 1 mM CaCl2, 1.2 mM KH2PO4, 1.2 mM MgSO4, 5 mM NaHCO3, 25 mM Hepes (pH 7.4), 3.5% BSA + 5.5 mM glucose) for 45 min with shaking at 37 °C[50], then centrifuged at $450 \times g$ for 5 min. The SVF pellet was washed twice with KRBA and once with DMEM complete medium. Then, the pellet was resuspended in 5 ml of DMEM complete medium and filtered over 70 μm filter adaptor. Filtered SVF was plated onto rat tail collagen-I (Invitrogen)-coated dish. For differentiation, confluent primary preadipocytes were differentiated with 50 nM insulin, 100 nM T3, 0.125 mM Indomethacin, 0.5 mM IBMX and 5 μM dexamethasone in DMEM/F12 media supplemented with 10% FBS for 2 days, followed by 4 days in medium supplemented with 50 nM insulin and 1 nM T3 with media change in between.

To confirm the inhibitory effect of scEMC10, differentiated primary adipocytes from either WT or KO mice were treated with either control or recombinant scEMC10 at 50 pg, 500 pg, 5000 pg or 5000 pg (heated inactivated) respectively for 24 h.

## RNA extraction and real time PCR

Total RNA was isolated from tissues and cells with the use of Trizol reagent (Invitrogen) and Direct-zol kit (Zymo). cDNA was prepared from 1 μg of total RNA using the High Capacity cDNA Reverse Transcription Kit (Invitrogen) with random hexamer primers, according to the manufacturer's instructions. The resulting cDNA was diluted 5-fold, and a 1.5 μl aliquot was used in a 6 μl PCR reaction (SYBR Green, Bio-Rad) containing primers at a concentration of 300 nM each. PCR reactions were run in triplicate and quantitated using the Applied Biosystems ViiA™7 Real-Time PCR system. Results were normalized to *TATA box binding protein* (*TBP*) expression and expressed as arbitrary units or fold change. Primer sequences listed in Supplementary Table 4.

## *scEMC10* mRNA expression in human visceral and subcutaneous adipose tissue

Adipose tissue sc*EMC10* mRNA expression was investigated in 240 donors of paired omental and SC adipose tissue samples (Supplementary Table 1). Adipose tissue was immediately frozen in liquid nitrogen after explantation. Human sc*EMC10* mRNA expression was measured by quantitative real-time RT-PCR in a fluorescent temperature cycler using the TaqMan assay, and fluorescence was detected on an ABI PRISM 7000 sequence detector (Applied Biosystems, Darmstadt, Germany). Primer sequences for human sc*EMC10* gene listed in Supplementary Table 4.

## Western blotting

Total cell or tissue lysates (20–50 μg) were subjected to SDS–PAGE followed by immunoblotting using specific antibodies and detection with chemiluminescence (Thermo Fisher). Multiple exposures were used to ascertain signal linearity. Images have been cropped for presentation[51].

## Co-immunoprecipitation

FLAG-hscEMC10, HA-PKA and HA-AKT1 were all cloned into the lentiviral vector pLEX-MCS-CMV-puro (Addgene, USA). Transfection

experiments were performed when the HEK293T cells were about 60–80% confluent, and cells were transfected with PEI reagents. Transfected 293T cells were lysed in EBC lysis buffer (50 mM Tris-HCl, pH 8.0, 120 mM NaCl, 0.5% Nonidet P-40) supplemented with protease inhibitors (Selleck Chemicals) and phosphatase inhibitors (Selleck Chemicals). For immunoprecipitation, cell lysates were incubated with anti-FLAG M2 agarose beads or anti-HA agarose beads for 2 h. Beads were then washed four times with NETN buffer (20 mM Tris-HCl, pH 8.0, 100 mM NaCl, 1 mM EDTA, and 0.5% Nonidet P-40). Then the precipitated samples were separated by 10% SDS-PAGE gel and blotted with indicated primary antibodies. Primary antibodies used for western blot analysis were as follows: anti-Flag (1:3000; F7425; Sigma Aldrich), anti-HA (1:3000; SC-7392; Santa Cruz Biotechnology), anti-PKA C-α (1:1000; D38C6; CST). Peroxidase-labeled anti-mouse (1:5000; P0217; DAKO) or anti-rabbit (1:5000; P0260; DAKO) IgG secondary antibody was used.

### scEMC10 translocation analysis
scEMC10 recombinant protein was labeled with FITC using a FITC conjugation kit (Sangon Biotech D601049). Cells were grown on glass coverslips for transfection or treatment as indicated, then treated with FITC-scEMC10 for 2 h, and recovered for the times indicated. The cells were fixed with 4% paraformaldehyde in PBS for 15 min at room temperature. Samples were rinsed three times with PBS (5 min for each wash). Nuclei were counterstained with 4,6-diamidino-2-phenylindole (DAPI) for 10 min. Cell membranes were counterstained with Octadecyl Rhodamine B Chloride (R18) (Sigma 83685). Coverslips were rinsed twice (3 min for each wash) with PBS and mounted onto slides using ProLong Gold Antifade reagent (Invitrogen). All images were obtained with the Leica TCS SP8 fluorescence microscope.

### In vitro kinase assay
Briefly, 2 μg recombinant GST-CREB N-terminal proteins were incubated with immunoprecipitated HA-PKA from transfected 293 T cells in the presence of 50 μM ATP and kinase reaction buffer (20 mM Tris-HCl, 50 mM NaCl, 10 mM KCl, 10 mM MgCl$_2$, 2 mM DTT, pH 7.5) at 30 °C for 40 min. The inhibitors H89-2HCl (10 μM), hscEMC10 protein (2 μg) and inactivation hscEMC10 protein (2 μg) were added as indicated. Reactions were stopped by adding in loading buffer and then analyzed by western blot.

### in vitro scEMC10 neutralization assay
DMEM containing 1 μg/ml of mouse scEMC10 protein were incubated with either 1 μg/ml of mouse anti-scEMC10 monoclonal antibody or control for 1 h before treating HeLa cells for 6 h. Samples were then lysed for western blotting for CREB phosphorylation (anti CREB-pS133, CST, #9198, 1:1000; anti CREB1, ABclonal, #A10826, 1:1000).

### Antibody neutralization in vivo
6-wk-old male C57BL/6J (Jackson Laboratory) mice were fed with HFD for 6-7 wk before injected with either IgG or monoclonal (3 mg/kg BW) anti-scEMC10 antibodies twice weekly.

### Reagents
Human insulin, isobutylmethylxanthine (IBMX), dexamethasone, T3, and indomethacin were purchased from Sigma-Aldrich. H89 and SB203580 were purchased from Tocris Bioscience. HY101120 was obtained from MedChem Express.

### Antibodies
Primary antibodies used for western blot analysis were as follows: anti-p-CREB (#9198; 1:1000; CST), anti-total CREB (#9197; 1:1000; CST), anti-p-P38MAPK (#4511; 1:1000; CST), anti-total P38MAPK (#8690; 1:1000; CST), anti-β-actin (#66009; 1:1000; Proteintech), anti-myc (#2278; 1:1000; CST), anti-αtubulin (#66031; 1:5000; Proteintech).

αTubulin (1:3000, Santa Cruz, Cat# SC-23948), Flag-tag rabbit (1:3000, Proteintech, Cat# 20543-1-AP), HA-tag mouse (1:3000, Santa Cruz, Cat# SC-7392), Phospho-CREB (Ser133) (1:1000, Cell Signaling Technology, Cat#9198), PKA C-α (1:1000, Cell Signaling Technology, Cat#5842), CREB (1:1000, Cell Signaling Technology, Cat#9197), GST (1:3000, Proteintech, Cat# 66001-1-AP). For Western blot, rabbit anti-EMC10 was used at 1:1000 dilution. Peroxidase-labeled anti-mouse (1:2000; #7076; CST) or anti-rabbit (1:2000; #7074; CST) IgG secondary antibody was used. Rabbit polyclonal and mouse monoclonal antibodies to EMC10 were raised against the recombinant protein of human scEMC10 (Phrenzer Biotechnology, Shanghai, China).

### Statistical analyses
**Anthropometric, metabolic characteristics and serum EMC10 concentration analyses.** All analyses were performed with Statistical Package for Social Sciences version 22.0 (SPSS, Chicago, IL, USA). GraphPad Prism software (version 7.0a for MAC; La Jolla, CA, USA) was used to plot histograms. Normally distributed data were expressed as means ± SD. Data that were not normally distributed, as determined using Kolmogorov–Smirnov test, were fourth root-transformed or lg-transformed before analysis and expressed as median with interquartile range. One-way ANOVAs with Fisher's *LSD post hoc* test were used to compare the differences among the three groups in cross sectional study. Proportions were compared by the Fisher's exact test. Student's paired *t* test was used for before and after comparison in observational studies. Pearson's correlation was used to evaluate the correlations between the fourth root-transformed serum EMC10 levels and clinical parameters. All two-tailed *P*-values <0.05 were considered significant.

**Serum EMC10 correlation analyses.** All statistical analyses were performed using the Statistical Package for Social Sciences software, version 20.0 for Windows (SPSS, Inc., Chicago, IL) and Prism (GraphPad, La Jolla, California). Data were presented as means and standard deviations (SDs) for continuous data, and categorical data were reported as frequency and percentage. Patients in the experimental and control groups were divided into subgroups based on their gender and BMI. Data with normal distributions were assessed by one-way ANOVA and t-test, in other cases, the Mann–Whitney U test was used. Correlations between serum EMC10 and other variables were assessed using Spearman's rank correlation analysis. All statistical analyses were two-tailed, and a *p*-value of <0.05 was considered to be statistically significant.

**Mouse models and in vitro studies analyses.** All data are presented as the mean ± S.E.M. (standard error of mean) and were analyzed by unpaired two-tailed Student's *t* test or analysis of variance, as appropriate. *P* < 0.05 was considered significant. Studies were performed on two or three independent cohorts and were performed on four to five mice per group unless specified. Sample size was determined using previous experiments on the characterization of the mouse model used in this study. Mice were randomized to treatment in a blinded manner whenever possible.

### Reporting summary
Further information on research design is available in the Nature Research Reporting Summary linked to this article.

## Data availability
The data supporting the findings from this study are available within the manuscript and its supplementary information. Emc10 knockout mouse model was generated by targeting DNA sequence (ENSMUSG00000008140). Source data underlying analyses in the human cohorts are available from the corresponding author upon reasonable request and are not publicly available due to patient

confidentiality / sensitive information. Source data underlying analyses in the mice cohorts are provided with this paper. Source data are provided with this paper.

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

## Acknowledgements

We thank Samuel M. Lockhart for his help in discussing the data and preparing the manuscript. The energy expenditure ANCOVA analysis done for this work was provided by the NIDDK Mouse Metabolic Phenotyping Centers (MMPC, www.mmpc.org) using their Energy Expenditure Analysis page (http://www.mmpc.org/shared/regression.aspx) and supported by grants DK076169 and DK115255. The authors also acknowledge the Research Open Access Publishing (ROAAP) Fund of the University of Illinois at Chicago for financial support towards the open access publishing fee for this article. The work was supported by R00 DK090210, R01 DK109015, University of Chicago DRTC (DK020595) Pilot & Feasibility award, Center for Society for Clinical and Translational Research Early Career Development Award and UIC Startup fund (C.W.L.); National Natural Science Foundation of China (No. 81070647, No. 81370936 and No. 81873645 to X.C.W., No. 81873853 to Q.D.), Shanghai Pujiang Program (No. 16PJ1401700) to X.C.W., Science and Technology Commission of Shanghai Municipality (No. 16140901200, No. 18140902100, and No. 22140902700) to X.C.W., China Diabetes Young Scientific Talent Research Project (2018-8) to X.C.W., National Basic Research Program (No. 2015CB943003) to Q. D. This work was further funded by the Deutsche Forschungsgemeinschaft: Collaborative Research Center SFB1052, project B1 (to M.B.) and supported by the EU/EFPIA Innovative Medicines Initiative Joint Undertaking (EMIF grant n° 115372). M.M was supported by American Heart Association Predoctoral Fellowship. R.N.K acknowledges support from RO1 DK067536.

## Author contributions

X.C.W., M.B., and C.W.L. conceived the project and experimental design. J.R.D. and Y.F.Y. analyzed human data. X.C.W., Y.L.L., G.F.Q., K.H.W., M.M., M.D.M., Z.H.Y., V.G., S.S.X., D.D.J., S.X.L., B.S., K.Y.C., Y.H.W., X.X.L., Q. M., D.M.G. and C.W.L. performed experiments and analyzed data. M.B. contributed human samples and supervised human serum and tissue expression analysis. J.C.C., L.N.Z., R.M.H., Q.D. and R.N.K. contributed samples and reagents. X.C.W. and C.W.L. wrote the paper. All authors discussed the results and commented on the manuscript.

## Competing interests

The authors declare no competing interests

## Additional information

[1]Department of Endocrinology, Huashan Hospital, Fudan University, Shanghai, China. [2]Department of Physiology & Biophysics, University of Illinois at Chicago, Chicago, IL, USA. [3]State Key Laboratory of Bioactive Substances and Functions of Natural Medicines, Institute of Materia Medica, Chinese Academy of Medical Sciences and Peking Union Medical College, Beijing, China. [4]State Key Laboratory of Cell Biology, Shanghai Institute of Biochemistry and Cell Biology, CAS Center for Excellence in Molecular Cell Science, Chinese Academy of Sciences, Shanghai, China. [5]University of Chinese Academy of Sciences, Beijing, China. [6]Department of Endocrinology and Metabolism, Renji Hospital, School of Medicine, Shanghai Jiao Tong University, Shanghai, China. [7]Research Division, Joslin Diabetes Center, Harvard Medical School, Boston, MA, USA. [8]Department of Medicine, Section of Endocrinology, Diabetes and Metabolism, University of Illinois at Chicago, Chicago, IL, USA. [9]NHC Key Laboratory of Hormones and Development, Tianjin Key Laboratory of Metabolic Diseases, Chu Hsien-I Memorial Hospital & Tianjin Institute of Endocrinology, Tianjin Medical University, Tianjin, China. [10]Department of Urology, Huashan Hospital, Fudan University, Shanghai, China. [11]Key Laboratory of Systems Health Science of Zhejiang Province, School of Life Science, Hangzhou Institute for Advanced Study, University of Chinese Academy of Sciences, Hangzhou, China. [12]Department of Medicine, University of Leipzig, Leipzig, Germany. [13]Present address: Department of Ophthalmology and Visual Sciences, University of Illinois at Chicago, Chicago, USA. [14]Present address: Department of Transplant Surgery, Mass General Hospital, Harvard Medical School, Boston, MA, USA. [15]These authors contributed equally: Xuanchun Wang, Yanliang Li, Guifen Qiang, Kaihua Wang. ✉e-mail: wangxch@fudan.edu.cn; cwliew@uic.edu

