## [Peer Review File · Nature Communications]

Title: Secreted EMC10 is upregulated in human obesity and its neutralizing antibody prevents diet-induced obesityREVIEWER COMMENTS

Reviewer #1 (Remarks to the Author):

This work provides a really thorough and detailed characterization of the metabolic function of scEMC10. Virtually nothing is known about the normal physiological function of this secreted protein (scEMC10). The authors have provided large amount of data to support their main hypothesis that scEMC10 is a novel secreted metabolic regulator.

Major strength of the study:

1. Novelty of the findings. This work will significantly advance the biological understanding of scEMC10 (a secreted protein with largely unknown function).
2. The potential impact and clinical relevance of the work is clear given the substantial amount of human data in the paper.
3. Most of the experiments are logical and well conducted.
4. The data are quite compelling, based on both loss- and gain-of-function mouse models. The physiological data in mice clearly support an important metabolic role for scEMC10.

Major weakness of the study:

1. The proposed mechanism (based on in vitro studies) is unclear. Conceptually, scEMC10 should go through the ER secretory pathway to be secreted since it contains a signal peptide. PKA, on the other hand, is an intracellular protein (kinase). As such, the biological relevance and significance of scEMC10 directly binding to the catalytic subunit of PKA to inhibit its enzymatic activity is not clear.

In summary, in spite of the weakness surrounding the proposed mechanism, this work is judged to be highly novel, impactful, and well done.

Reviewer #2 (Remarks to the Author):

Wang et al's new manuscript "scEMC10, a novel circulating inhibitor..." is well written and describes a novel role for this circulating protein in promoting obesity and associated metabolic syndrome. Authors describe with a systematic approach, by providing compelling evidence in mice obese models as well as through correlation of high levels of circulating scEMC10 in human obese subjects compared to age matched lean control group. The correlation of circulating EMC10 to the excess adiposity and insulin resistance is very striking. and Authors used whole body KO mice as well as transgenic overexpression, through AAV system, of this novel circulating protein factor to demonstrate how scEMC10 promote mice model for obesity.

Authors also demonstrated such effect is not mediated through usual suspect of food intake or activity, but through inhibiting adipocyte thermogenesis. In addition to KO mice they have also used a pharmacological tool, neutralizing anti-scEMC10 ab which strengthen the arguments that scEMC10's role in promoting obesity. They have attempted to understand the MOA of this novel protein by using ex

vivo generated tissues and cells to link their role in regulation of adipocyte thermogenic capacity through inhibition of PKA/CREB axis and the scEMC bind to PKA. All the experiments are performed to arrive at a logical conclusion and carried out thoughtfully.

I am happy to approve this manuscript for publication once addressed following comments.

Major Points:

1. It is interesting that thermogenesis related genes are upregulated in KO mice in both HFD as well as Low fat diet fed mice (Figure 4F & 4H). Authors have shown the metabolic cage studies to look at energy expenditure (Fig 4A & B) only for the HFD condition and not the CD fed mice. It will be informative for authors to include CD diet metabolic cage data as well. This is important mainly due to the fact that scEMC10 is shown to mediate the protective effect from HFD feeding, is due to adipocyte thermogenesis and not food intake or activity.

2. Continuing my argument in the same line above (to focus more on whole body physiological effect), It would be important to show that scEMC10 mediate its protective effect predominantly at HFD/condition by subjecting the mice to metabolic cage in control diet and switch to HFD (both KO and WT). This will be an elegant way to demonstrate, how acutely lack or presence of scEMC10 contribute to energy expenditure.

3. While authors very nicely used AAV based over expression of EMC10 to demonstrate that this protein is the main driver of obesity (Fig 3J-M), another important experiment is to subject the mice to metabolic cages, in this acute setting, especially before and after AAV treatment. This will substantiate the authors main claim.

4. Authors very nicely demonstrate not only using genetically ablated mice (of EMC10), but also successfully raised neutralizing ab to demonstrate in the in vivo setting. The neutralizing effect is acute (Figure 6A&6B). Its not very clear in the text as well as legend, when exactly the GTT/ITT and serum markers are looked in to. Similar to earlier comment, on metabolic cage data, it would be elegant perform the experiment before and after 4c2 dosing, to see how acutely the metabolic shift happen.

Minor points:

1. In the metabolic cage studies (Figure 4A), its surprising not to see the regular circadian pattern seen in mice (between Dark and Light cycle, similar to their own data (Fig 6K). Minor point: The scales (Y axis) used in the metabolic cage data graphs (between Fig 4A and Fig 6K need to be similar to have consistency in data representation).

2. The chow-fed cohort (Line 228) should be changed to Low-fat diet fed mice, to be consistent with the experiment and fig legend (sup Fig 4G) or use consistent terminology throughout. (line 303, also used

CD-fed mice for Fig 4H).

3. We observed EMC10 KO improved (Line 420) should be changed to 'protected steatosis..'

Reviewer #3 (Remarks to the Author):

In submitted paper entitled "scEMC10, a novel circulating inhibitor of adipocyte thermogenesis, is upregulated in human obesity and its neutralizing antibody prevents diet-induced obesity", Xuanchun et al. identify scEMC10 is upregulated in humans with obesity and is positively associated with insulin resistance. And they identify scEMC10 as a circulating inhibitor of thermogenesis via PKA signaling.

Their hypothesis is supported by a genetic and a pharmacological mouse model. Even though the study is interesting and introduces a novel mechanism, there are several weak points that need to be addressed.

Major points:

1. In Figure 2-B, E, why EMC10 concentration is quite different Baseline level?

In Figure 2B looks EMC10 is over 100 ng/ml, contrast to this Figure 2E looks EMC10 is around 10 ng/ml. In other words, why EMC 10 level of Bariatric surgery groups differ from diet groups?

2. scEMC10 is secreted many tissues, but which tissues are not clear main source in vivo. Authors used total KO mice, so it is very difficult address that question. Do the authors have adipocyte-specific KO mice data? Such data strongly support their conclusion.

3. Figure 4, KO mice showed increased VO₂ and RQ. Muscle is also important thermogenic organ, authors should be evaluated muscle shivering. Increased RQ indicate glucose metabolism is enhanced, do authors have any glucose metabolism data?

4. Figure 5, KO primary adipocytes increased Ucp1, Pgc1a- what is the mechanism of increased thermogenic adipocytes? How about Mitochondrial biogenesis or cell differentiation, do author have the data?

We thank the Reviewers for their constructive critiques. We believe the large number of additional experiments and changes in the text have made the manuscript stronger.

Please note for clarity our responses are in ***italics in red and underlined***:

Reviewer #1 (Remarks to the Author):

This work provides a really thorough and detailed characterization of the metabolic function of scEMC10. Virtually nothing is known about the normal physiological function of this secreted protein (scEMC10). The authors have provided large amount of data to support their main hypothesis that scEMC10 is a novel secreted metabolic regulator.

Major strength of the study:

1. Novelty of the findings. This work will significantly advance the biological understanding of scEMC10 (a secreted protein with largely unknown function).
2. The potential impact and clinical relevance of the work is clear given the substantial amount of human data in the paper.
3. Most of the experiments are logical and well conducted.
4. The data are quite compelling, based on both loss- and gain-of-function mouse models. The physiological data in mice clearly support an important metabolic role for scEMC10.

Major weakness of the study:

1. The proposed mechanism (based on in vitro studies) is unclear. Conceptually, scEMC10 should go through the ER secretory pathway to be secreted since it contains a signal peptide. PKA, on the other hand, is an intracellular protein (kinase). As such, the biological relevance and significance of scEMC10 directly binding to the catalytic subunit of PKA to inhibit its enzymatic activity is not clear.

In summary, in spite of the weakness surrounding the proposed mechanism, this work is judged to be highly novel, impactful, and well done.

Indeed, our data in the original submission did not clearly show the mechanism by which extracellular scEMC10 interacts with intracellular PKA. Since we have detected the secretion of scEMC10 (Supplementary Fig 1) and have seen the extracellularly-added, epitope-tagged scEMC10 in the cytosol fraction of our subcellular fractionation assay (preliminary data not shown), we initially hypothesized that scEMC10 is transported into the cell to exert its functions. To confirm our hypothesis, we labeled the recombinant scEMC10 with FITC and then treated Hela cells on coverslips for 2 h. After treatment, cells were washed with PBS for 3 times before imaging. As shown in our data (Fig 6D and Reviewer Fig 1), we detected the labeled scEMC10 within the cells. Our current data support our hypothesis that scEMC10 is transported into cell to mediate its metabolic effects.

Reviewer #2 (Remarks to the Author):

Wang et al's new manuscript "scEMC10, a novel circulating inhibitor..." is well written and describes a novel role for this circulating protein in promoting obesity and associated metabolic syndrome. Authors describe with a systematic approach, by providing compelling evidence in mice obese models as well as through correlation of high levels of circulating scEMC10 in human obese subjects compared to age matched lean control group. The correlation of circulating EMC10 to the excess adiposity and insulin resistance is very striking. and Authors used whole body KO mice as well as transgenic overexpression, through AAV system, of this novel circulating protein factor to demonstrate how scEMC10 promote mice model for obesity.

Authors also demonstrated such effect is not mediated through usual suspect of food intake or activity, but through inhibiting adipocyte thermogenesis. In addition to KO mice they have also used a pharmacological tool, neutralizing anti-scEMC10 ab which strengthen the arguments that scEMC10's role in promoting obesity. They have attempted to understand the MOA of this novel protein by using ex vivo generated tissues and cells to link their role in regulation of adipocyte thermogenic capacity through inhibition of PKA/CREB axis and the scEMC bind to PKA. All the experiments are performed to arrive at a logical conclusion and carried out thoughtfully.

I am happy to approve this manuscript for publication once addressed following comments.

Major Points:

1. It is interesting that thermogenesis related genes are upregulated in KO mice in both HFD as well as Low fat diet fed mice (Figure 4F & 4H). Authors have shown the metabolic cage studies to look at energy expenditure (Fig 4A & B) only for the HFD condition and not the CD fed mice. It will be informative for authors to include CD diet metabolic cage data as well. This is important mainly due to the fact that scEMC10 is shown to mediate the protective effect from HFD feeding, is due to adipocyte thermogenesis and not food intake or activity.

Consistent with our body weight data (Supplementary Fig 4), the KO mice fed low-fat diet had a trend towards greater energy expenditure but no significant differences in our metabolic cage studies (Supplementary Fig 8A, Reviewer Fig 2), despite significantly increased thermogenesis-related genes in the adipose tissues of the KO mice. This discrepancy could potentially be due to the fact that whole-body energy expenditure is contributed by a number of different metabolic organs. In our case, since the dietary and metabolic cage studies were conducted at room temperature which is known to cause mild muscle shivering, this could potentially contribute to what we have detected in our metabolic cages.

2. Continuing my argument in the same line above (to focus more on whole body physiological effect), It would be important to show that scEMC10 mediate its protective effect predominantly at HFD/condition by subjecting the mice to metabolic cage in control diet and switch to HFD (both KO and WT). This will be an elegant way to demonstrate, how acutely lack or presence of scEMC10 contribute to energy expenditure.

We would like to thank the reviewer for suggesting an elegant approach to potentially dissect the acute contribution of scEMC10 on energy expenditure. Following reviewer recommendation, we subjected both WT and KO mice to dietary switch while being monitored in the metabolic cage. As shown in Reviewer Fig 3, we observed no difference in the VO₂, VCO₂, and energy expenditure between WT and KO mice before and after LFD/HFD switch. However, based on the RQ data, we confirmed that the experiment was conducted successfully. Our data showed that upon switching to HFD, the RQ value for both the WT and KO mice reduces instantaneously, which suggests these animals consumed more fat as energy. In addition, we also notice that under the LFD feeding, the KO mice have the tendency to use more carbohydrate compared to WT (higher RQ value) but upon switching to HFD, the KO mice showed a biphasic change. Under HFD, the KO mice tend to have a lower RQ value during the light phase but higher RQ value during the dark phase. The implication of these changes is currently unknown; however, differences in substrate utilization affected by the activity of metabolic organs could contribute to body weight changes in the long run.

3. While authors very nicely used AAV based over expression of EMC10 to demonstrate that this protein is the main driver of obesity (Fig 3J-M), another important experiment is to subject the mice to metabolic cages, in this acute setting, especially before and after AAV treatment. This will substantiate the authors main claim.

Based on reviewer recommendation, right after we measured the baseline metabolic profile of the WT mice, we injected the mice with either AAV-Laz control or AAV-hscEMC10. In order to allow for the expression of the injected transgene, based on our prior experience, we waited for 10 days before subjecting the mice to metabolic cages measurement again. As shown in Supplementary Fig 8B and 8C (Reviewer Fig 4), we observed that overexpression of scEMC10 significantly downregulates VO₂, VCO₂, and energy expenditure of the AAV-hscEMC10 mice.

4. Authors very nicely demonstrate not only using genetically ablated mice (of EMC10), but also successfully raised neutralizing ab to demonstrate in the in vivo setting. The neutralizing effect is acute (Figure 6A&6B). Its not very clear in the text as well as legend, when exactly the GTT/ITT and serum markers are looked in to. Similar to earlier comment, on metabolic cage data, it would be elegant perform the experiment before and after 4c2 dosing, to see how acutely the metabolic shift happen.

GTT, ITT, and serum markers were done right after the treatment (2-3 wk from the start of treatment) which is

between 9-10 weeks after the onset of high fat diet (Figure 7a). This information has been added to the text and figure legends. We have replaced Figure 7K with graph showing data before and after 4C2 dosing.

Minor points:

1. In the metabolic cage studies (Figure 4A), its surprising not to see the regular circadian pattern seen in mice (between Dark and Light cycle, similar to their own data (Fig 6K). Minor point: The scales (Y axis) used in the metabolic cage data graphs (between Fig 4A and Fig 6K need to be similar to have consistency in data representation).

Notable differences in the metabolic cages data between Fig 4A and 6K is due to the fact that data presented in 4A was done earlier with the Columbus CLAMS system and the data in 7K was done using our newly acquired Sable Promethion Metabolic Monitoring System. Our new system was able to show higher resolution data. To ensure consistency, we have repeated the experiment and replaced Fig 4A with new data.

2. The chow-fed cohort (Line 228) should be changed to Low-fat diet fed mice, to be consistent with the experiment and fig legend (sup Fig 4G) or use consistent terminology throughout. (line 303, also used CD-fed mice for Fig 4H).

Corrected. Please note the WT and KO experiments were done with LFD and HFD whereas the hscEMC10 overexpression experiments were done with chow and high-fat diets.

3. We observed EMC10 KO improved (Line 420) should be changed to 'protected steatosis..'

Changed.

Reviewer #3 (Remarks to the Author):

In submitted paper entitled "scEMC10, a novel circulating inhibitor of adipocyte thermogenesis, is upregulated in human obesity and its neutralizing antibody prevents diet-induced obesity", Xuanchun et al. identify scEMC10 is upregulated in humans with obesity and is positively associated with insulin resistance. And they identify scEMC10 as a circulating inhibitor of thermogenesis via PKA signaling.

Their hypothesis is supported by a genetic and a pharmacological mouse model. Even though the study is interesting and introduces a novel mechanism, there are several weak points that need to be addressed.

Major points:

1. In Figure 2-B, E, why EMC10 concentration is quite different Baseline level?

In Figure 2B looks EMC10 is over 100 ng/ml, contrast to this Figure 2E looks EMC10 is around 10 ng/ml.

In other words, why EMC 10 level of Bariatric surgery groups differ from diet groups?

Main reason to cause the discrepancy in the baseline value of EMC10 concentration is because the mean BMI for the bariatric surgery cohort is 50.91 kg/m² (Supplementary table 3) whereas the diet/exercise cohort is 35 kg/m² and according to our data, serum concentration of EMC10 doesn't increase linearly with higher BMI (Reviewer Fig 5). Therefore, it is reasonable to see different EMC10 baseline levels for the Bariatric surgery and the diet groups.

2. scEMC10 is secreted many tissues, but which tissues are not clear main source in vivo. Authors used total KO mice, so it is very difficult address that question. Do the authors have adipocyte-specific KO mice data? Such data strongly support their conclusion.

We are currently generating the tissue specific KO mice, we hope to address this question in our future studies. Based on our current data, the source of EMC10 might not be critical as long as circulating EMC10

could be neutralized for the therapeutic standpoint. But yes, it will be important to determine which tissues secrete EMC10 to understand its regulation under different physiological and patho-physiological conditions.

3. Figure4, KO mice showed increased VO₂ and RQ. Muscle is also important thermogenic organ, authors should be evaluated muscle shivering. Increased RQ indicate glucose metabolism is enhanced, do authors have any glucose metabolism data?

We know that under thermoneutral conditions, muscle shivering will be at a minimum. Our data showed that even under the thermoneutral condition, ablation of EMC10 was still able to protect mice from HFD-induced obesity. Therefore, based on our current data, we expect that muscle shivering might not significantly contribute to our observed phenotype in the EMC10 KO mice. Increase in RQ in the KO mice is supported by improvement in the glucose tolerance as shown by IPGTT data in Fig 3D.

4. Figure5, KO primary adipocytes increased Ucp1, Pgc1a- what is the mechanism of increased thermogenic adipocytes?

How about Mitochondrial biogenesis or cell differentiation, do author have the data?

To address the reviewer's concern, we examined the expression of additional mitochondrial biogenesis (Nrf1, Nrf2), mitochondrial fusion (Mfn 1, Mfn 2, Opa1), fission (Drp1, Fis1) markers, and adipocyte differentiation (C/EBPa, C/EBPb, C/EBPd, and PPARg2) markers in WT and KO differentiated primary adipocytes. Consistent with the increase in Pgc1a, which is an upstream regulator of NRF genes, we also observed increase in the expression of Nrf1 and Nrf2 in the KO adipocyte (Reviewer Fig 6); however, we found no change in the expression of mitochondrial fusion, fission and adipocyte differentiation markers (Reviewer Fig 6). Our data suggest that potential increased mitochondria biogenesis in the KO adipocyte is largely mediated by the increase in Pgc1a. We are happy to include this data in our final manuscript if recommended by the reviewer.

Figure 1

Figure 1. scEMC10 is transported into the cell.

Immunofluorescence analysis of HeLa cells treated with either scEMC10 labeled with FITC or control. Nuclei were stained blue with 4',6-diamidino-2-phenylindole (DAPI), and cell membranes were stained red with Octadecyl Rhodamine B Chloride (R18).

Figure 2

Figure 2. Emc10 KO mice fed low-fat diet had a trend towards greater energy expenditure.

Heat production (left), oxygen consumption (VO₂) (middle) and carbon dioxide production (VCO₂) (right) were analyzed by indirect calorimetry for 48h in WT (black circle) or KO (red square) mice under low fat diet (LFD) (n=8 per group).

Figure 3

Figure 3. No difference in energy expenditure between WT and KO mice before and after LFD/HFD switch.

Oxygen consumption (VO_2), carbon dioxide production (VCO_2), Heat production, and respiratory rate (RQ) were analyzed by indirect calorimetry for 96h in WT (black circle) or KO (red square) mice before and after LFD/HFD switch ($n=6$ per group).

Figure 4

Figure 4. Energy expenditure before and after overexpression of hscEMC10.

(A) Heat production, oxygen consumption (VO₂) and carbon dioxide production (VCO₂) were analyzed by indirect calorimetry for 40h in B6 mice before AAV injection (n=6 per group). (B) 10 days after either AAV-LacZ control (black circle) or AAV-hscEMC10 (red square) injection, heat production, oxygen consumption (VO₂) and carbon dioxide production (VCO₂) were analyzed by indirect calorimetry for 40h (n=6 per group).

Figure 5

Figure 5. Serum concentration of EMC10 doesn't increase linearly with higher BMI
The correlation between serum EMC10 and BMI in a Caucasian cohort.

Figure 6

Figure 6. No change in the expression of mitochondrial fusion, fission and adipocyte differentiation markers.

The expression of mitochondrial biogenesis (Nrf1, Nrf2), mitochondrial fusion (Mfn 1, Mfn 2, Opa1), fission (Drp1, Fis1) markers, and adipocyte differentiation (C/EBP α , C/EBP β , C/EBP δ , and PPAR γ 2) markers in WT (black) and KO (red) differentiated primary adipocytes (n= 6 per group). All data are presented as mean +/- SEM. *, p<0.05; **, p<0.01.

REVIEWERS' COMMENTS

Reviewer #1 (Remarks to the Author):

The authors have successfully addressed my concerns and comments.
No further issue.

Reviewer #2 (Remarks to the Author):

Thank you the authors for addressing the questions and take corrective action and incorporated in the manuscript. I am happy about the way authors addressed the concerns.

I am in full agreement to publish this manuscript as the revised version meets the quality of "nature communications".

all the best to authors

Reviewer #3 (Remarks to the Author):

I think the revised manuscript will be accepted. I am satisfied with their efforts and revision.